# Comparative analysis of the complete chloroplast genome of Papaveraceae to identify rearrangements within the *Corydalis* chloroplast genome

**Sang-Chul Kim, Young-Ho Ha, Beom Kyun Park, Ju Eun Jang, Eun Su Kang, Young-Soo Kim, Tae-Hee Kimspe, Hyuk-Jin Kim***

Division of Forest Biodiversity, Korea National Arboretum, Pocheon, Republic of Korea

* jins77@korea.kr

**Data Availability Statement:** All of our data is registered with NCBI (https://www.ncbi.nlm.nih.gov/). S1, S2, and S3 Tables are all recorded.

## Abstract

Chloroplast genomes are valuable for inferring evolutionary relationships. We report the complete chloroplast genomes of 36 *Corydalis* spp. and one *Fumaria* species. We compared these genomes with 22 other taxa and investigated the genome structure, gene content, and evolutionary dynamics of the chloroplast genomes of 58 species, explored the structure, size, repeat sequences, and divergent hotspots of these genomes, conducted phylogenetic analysis, and identified nine types of chloroplast genome structures among *Corydalis* spp. The *ndh* gene family suffered inversion and rearrangement or was lost or pseudogenized throughout the chloroplast genomes of various *Corydalis* species. Analysis of five protein-coding genes revealed simple sequence repeats and repetitive sequences that can be potential molecular markers for species identification. Phylogenetic analysis revealed three subgenera in *Corydalis*. Subgenera *Cremnocapnos* and *Sophorocapnos* represented the Type 2 and 3 genome structures, respectively. Subgenus *Corydalis* included all types except type 3, suggesting that chloroplast genome structural diversity increased during its differentiation. Despite the explosive diversification of this subgenus, most endemic species collected from the Korean Peninsula shared only one type of genome structure, suggesting recent divergence. These findings will greatly improve our understanding of the chloroplast genome of *Corydalis* and may help develop effective molecular markers.

## 1. Introduction

Chloroplasts are organelles responsible for photosynthesis and oxygen release and are essential for plant survival. In addition, chloroplasts also play an important role in taxonomic and evolutionary studies on plants [1, 2]. The chloroplast genome is known to be more conserved than the nuclear or mitochondrial genomes in terms of genetic structure, gene content, and the nucleotide sequence [3–5]. Because of its highly conserved and non-recombinant nature, the

**Funding:** This research was funded by Scientific Research Grants of the Korea National Arboretum (grant number KNA1-1-13, 14–1). Also, the funders had no role in study design, data collection and analysis, decision to publish, or preparation of the manuscript.

**Competing interests:** The authors have declared that no competing interests exist.

chloroplast genome is a very useful genetic resource for inferring evolutionary relationships at various taxonomic levels [6]. The typical chloroplast genome of angiosperms is a circular molecule of double-stranded DNA (length, 120–170 kb) that encodes up to 101–118 genes, including 66–82 protein-coding genes, 29–32 tRNA genes, and 4 rRNA genes [7]. The chloroplast genome is a quadrangular structure composed of a small single-copy region (SSC) and a large single-copy (LSC) region linked by a pair of inverted repeats (IRa and IRb) [8]. Recent technological developments have made it simpler and cheaper to analyze the complete chloroplast genome, and the chloroplast genomes of several species have been reported to date.

Major genomic structural changes—such as gene loss, large inversions, and contraction or expansion of the IR region—are frequently reported in the plastid genomes of certain plant lineages. For example, certain conifers [9, 10] and some Fabaceae plants [11] have lost one IR region. Some Pinaceae species have been shown to possess smaller IRs with a size of less than 1 kb [12, 13], and some Ericaceae species exhibit a greatly reduced SSC region [14–17] due to IR extension. In *Lamprocapnos spectabilis* (L.) Fukuhara, the IR region is extended to the LSC and SSC regions, and these boundary shifts are accompanied by five inversion regions [18]. Reversals in the chloroplast genome not only allow us to understand its evolutionary patterns [19, 20] but also provide markers for identifying subgenera [21]. Recent phylogenomic studies in Ranunculaceae suggest that the chloroplast genome exhibits distinct evolutionary patterns depending on the phylogenetic lineage [20, 22]. Xu and Wang [23] reported unusual large-scale rearrangements in four newly reported chloroplast genomes of *Corydalis* (Papaveraceae; encompassing three subgenera) and found evidence of IR expansion, SSC contraction, and relocation. Structural variations are important regulators of key characteristics representative of a specific group [21, 23] or species [24]. Therefore, inferring structural variations in the chloroplast genomes of various species and their related genera (for example, *Corydalis*) is an interesting research topic, and the results may provide unexpected insights into their evolutionary lineage.

The genus *Corydalis* DC. (Papaveraceae: Fumarioideae) comprises approximately 465 taxa with worldwide distribution [25, 26]. This genus is clearly distinguished from other genera by its characteristic features, which include symmetrical flowers, distinct structures of the stamens and petals, and the persistence of the style after fruit maturation [27]. The genus is naturally distributed in the Northern Hemisphere, and is mainly found in temperate regions. However, *Corydalis* is especially prevalent in China and Tibet—regions that may have undergone intensive and rapid species differentiation due to extreme geographical changes and geological upheaval in the past [23, 28–30]. A recent study estimated the differentiation period of this genus to be 44 Mya (the Paleogene) based on molecular phylogenetic analyses of the *matK* gene, *trnL* intron, *trnL-F* intergenic spacer region, *trnG* intron, *rps16* intron, 26S gene, and internal transcribed spacer (ITS) region. The Himalayan region has been estimated to be the center of this range expansion [29, 31]. A total of 35 species of *Corydalis* have been reported in South Korea (National Standard Plant List of the Korea National Arboretum, http://www. nature.go.kr/kpni/SubIndex.do, accessed on July 13, 2022). Moreover, 14 species are endemic to the Korean Peninsula, including some recently described species [32–34]. Although the genomes of endemic species are known to be important genetic resources, the sequences of *Corydalis* spp. found in the Korean Peninsula have not been analyzed to date.

Genus *Corydalis* is one of the most taxonomically complex plant taxa, and the species included within it are very difficult to discriminate [35]. The first molecular phylogenetic study of this genus was conducted by Lidén et al. [36], who analyzed the *rps16* locus and morphological characteristics of plants in this genus. Based on their results, the researchers divided genus *Corydalis* into three subgenera: *Chremnocapnos*, *Sophorocapnos*, and *Corydalis*. Following this, Wang [37] analyzed two chloroplast regions (*matK* and *rps16*) and used palynology to

divide genus *Corydalis* into five subgenera: *Chremnocapnos*, *Sophorocapnos*, *Corydalis*, *Rapiferae*, and *Fasciculatae*. In a recent study, Ren et al. [35] analyzed two nuclear ITS (ITS1, ITS2) and one region of chloroplast DNA (*matK*) and reported that their results supported the classification suggested by Lidén et al. [36]. To avoid confusion, this study follows the classification system suggested by Lidén et al. [36].

The chloroplast genomes of 12 *Corydalis* spp. have been published to date, with most being published in China, where this genus is mainly distributed [16, 23, 38–41]. In this study, we compared the chloroplast genome sequences of *Corydalis* spp. to infer their structural variations and phylogenetic relationships. *Fumaria officinalis* L. (Papaveraceae: Fumarioideae) is a phylogenetically close relative of *Corydalis* [42]. Although this species is reportedly native to Europe [43], it is found worldwide due to its high dispersal ability. *Fumaria officinalis* has been observed on Jeju Island, located in the south of Korea, and it is designated as an invasive species at present (http://www.nature.go.kr/ and https://kias.nie.re.kr/, Accessed on July 13, 2022). To date, only *Corydalis* spp. and *Lamprocapnos* spp. have been analyzed in the subfamily Fumarioideae [16, 18, 23, 38–41, 44]. Therefore, to explore the diverse genetic structures of *Corydalis* and its close relatives, we included *F. officinalis* in our analysis.

This study had the following objectives. First, we reported the complete chloroplast genomes of 24 *Corydalis* spp., including 9 species endemic to Korea, 11 species collected from the Korean Peninsula, and 2 species collected from Kyrgyzstan. Second, we reported the complete chloroplast genome of *Fumaria* (a close relative of *Corydalis*), which has not been reported to date. Third, we constructed the chloroplast genomes of 11 *Corydalis* spp. for comparative genome analysis using the Short Reads Archive (SRA) database of the National Center for Biotechnology Information (NCBI). Finally, we analyzed the evolutionary patterns of the chloroplast genome of *Corydalis* by comparatively analyzing the completed chloroplast genomes of different species.

## 2. Materials and methods

### 2.1. Plant sampling and sequencing

*We collected 25* Corydalis spp., including 22 species distributed in Korea and 2 species from Kyrgyzstan. One F. officinalis collected from Jeju Island in the Korean Peninsula was also included in our analysis. Voucher specimens have been deposited in the Herbarium of the Korea National Arboretum (KH; S1 Table). No permit was required to take the samples that were not on the list of national key protected plants. Total genomic DNA was extracted using the DNeasy Plant Mini Kit (Qiagen Inc., Valencia, CA, USA) according to the manufacturer's instructions. The concentration and quality of the extracted DNA were determined with a NanoDrop 2000 system (Thermo Fisher Inc., Waltham, MA, USA) and electrophoresis us-ing a 1% agarose gel. Illumina paired-end libraries were constructed and sequenced on the MiSeq platform (Macrogen Inc., Seoul, Korea). In addition, the raw sequence data of 11 species were downloaded from the NCBI SRA database and used for analysis (S2 Table).

### 2.2. Chloroplast genome assembly and annotation

The chloroplast genome sequence was assembled into the scaffold using GetOrganelle v1.7.6.1 [45]. The genome sequence was completed with Geneious Prime® 2022.1.1 [46] and annotated using GeSeq [47]. The chloroplast genomes were compared, verified, and modified with Geneious Prime®2022.1.1. Transfer RNA (tRNA) sequences were annotated with the tRNAscan-SE software [48]. The boundaries of genes, introns, and coding regions were identified through comparisons with reference sequences. Finally, we mapped the circular chloroplast genome using OrganellarGenomeDRAW (OGDRAW) [49].

## 2.3. Structural analysis, analysis of relative synonymous codon usage, and repeat analysis

The genome structure of the analyzed chloroplasts (total genome length; length of the LSC, SSC, and IR; guanine-cytosine (GC) content; number of genes) was calculated using Geneious Prime®2022.1.1. Codon usage bias was analyzed using MEGA X [50]. SSRs (mononucleotides, dinucleotides, trinucleotides, tetranucleotides, pentanucleotides, and hexanucleotides; set to 10, 5, 4, 3, 3, and 3, respectively) and long repeat sequences (forward, palindromic, reverse, and complement repeats; set to a minimum repeat size of 30 bp and Hamming Distance 3) were identified using the online MISA tool [51] and REPuter [52], respectively.

## 2.4. Comparison of whole genomes

The whole genome alignments of taxa representing each type of chloroplast genome were compared with the mVISTA program in Shuffle-LAGAN mode and LAGAN mode with *Euptelea pleiosperma* Hook. f. & Thomson as a reference. In addition, the whole genome alignment of each taxon representing each type of chloroplast genome was compared using the mVISTA program in LAGAN mode [53].

## 2.5. Identification of the hotspots of divergence

DNA polymorphism was analyzed using DNA Sequence Polymorphism (DnaSP) v6 [54] to calculate the nucleotide diversity (*Pi*) of genes and to identify highly variable genes. Chloroplast genome sequences were aligned using MAFFT implemented in Geneious Prime® 2022.1.1.

## 2.6. Phylogenetic analyses

Phylogenetic analyses were performed using 58 taxa, including 36 newly identified species of Papaveraceae in this study, 21 other species of Papaveraceae, and *E. pleiosperma* (Eupteleaceae; used as an outgroup) (S3 Table). Phylogenetic analysis was performed using 77 protein-coding genes (CDSs). The phylogenetic trees of complete chloroplast genomes were constructed using the maximum likelihood (ML) and Bayesian inference (BI) methods.

The ModelFinder software [55] was used to detect the most suitable model for molecular phylogenetic analysis with ML (corrected Akaike's information criterion) and BI (Bayesian information criterion). The ML-based phylogenetic analysis was performed in Phylosuite using IQ-TREE [56, 57] under the TVM+F+R5 model for 50,000 ultrafast bootstraps [58]. The BI-based phylogeny was inferred using MrBayes 3.2.6 [57, 59] under the GTR+F+I+G4 model (2 parallel runs; 2,000,000 generations), in which the initial 25% of sampled data were discarded as burn-in.

# 3. Results

## 3.1. General features of Papaveraceae

All 58 chloroplast genomes—including 57 species of Papaveraceae and one species of Eupteleaceae as the outgroup—had a circular quadripartite structure with two IRs, one LSC, and one SSC. However, a total of nine types of structures were identified overall (Fig 1). The chloroplast genomes varied in size from 149,919 bp (*C. mucronifera*) to 200,923 bp (*C. chrysosphaera*), the LSC from 82,391 bp (*C. edulis*) to 98,393 bp (*C. fangshanensis*), the SSC from 72 bp (*C. intermedia*) to 25,869 bp (*C. ledebouriana*), and the IR from 22,777 bp (*C. pauciovulata*) to 54,930 bp (*C. chrysosphaera*). The average GC content of the chloroplast genome was 40.2%, and the

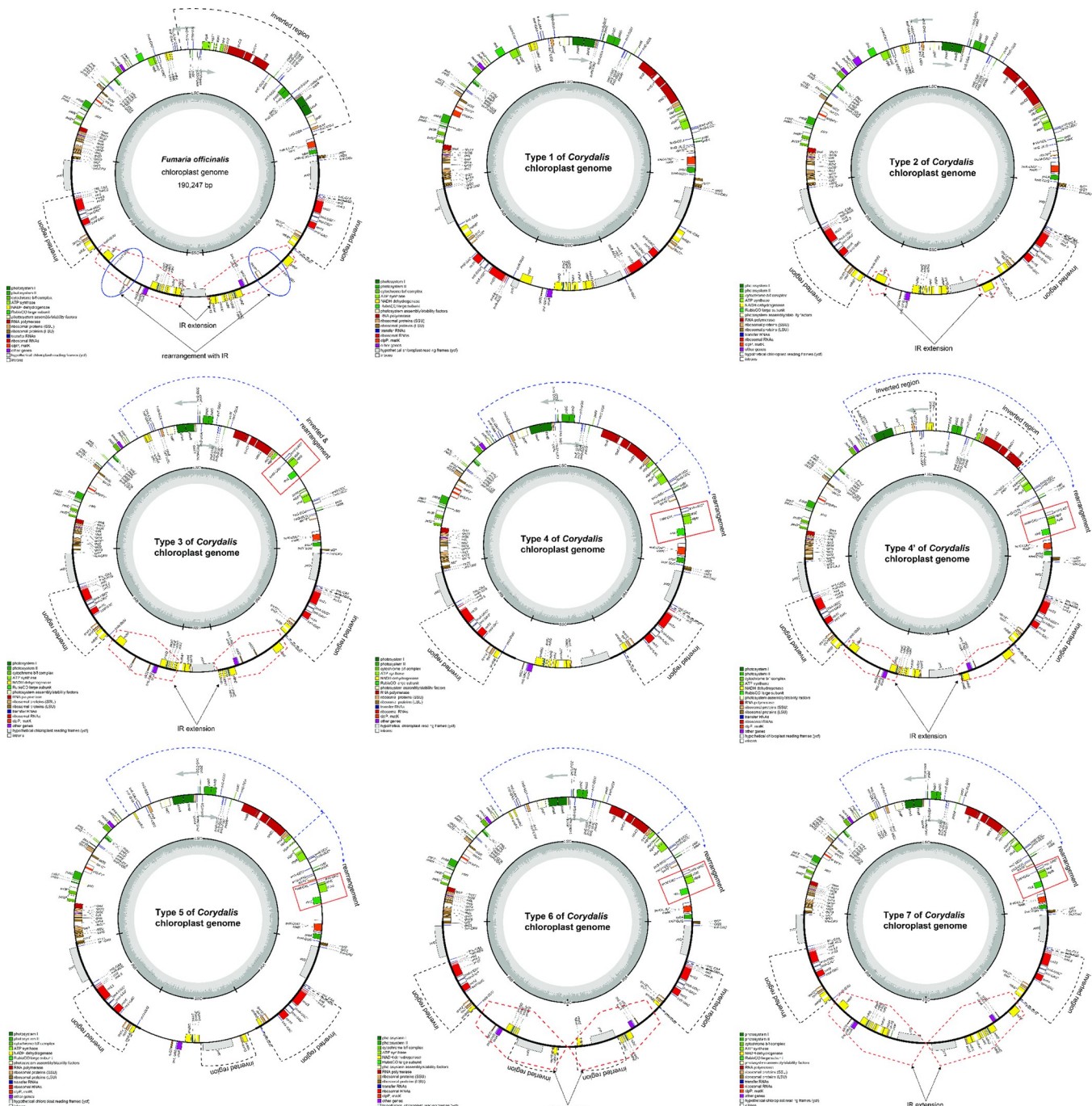

**Fig 1. Nine types of chloroplast genomes mapped in Papaveraceae species.** Genes shown outside the circle are transcribed in the counter-clockwise direction, whereas those inside the circle are transcribed in the clockwise direction. The colored bars indicate genes belonging to different functional groups. The darker gray region inside the inner circle denotes GC content, whereas the lighter gray region corresponds to the AT content of the genome. The ψ signifies pseudogenes.

total GC content ranged from 38.6% (*E. pleiosperma* and *Macleaya cordata*) to 41.5% (*C. trisecta* and *C. pauciovulata*). The GC content of the LSC ranged from 37% (*Macleaya cordata*) to 39.9% (*C. ternate* and *C. inopinata*), the IR from 40.1% (*L. spectabilis*) to 46.5% (*C. pauciovulata*), and the SSC from 15.6% (*C. namdoensis*) to 39.6% (*C. pauciovulata*; Table 1).

**Table 1. Characteristics of the chloroplast genomes of 57 species of Papaveraceae and one species of Eupteleaceae (outgroup).**

| Scientific name | Accession | Total length | GC (%) | LSC | GC (%) | IRs | GC (%) | SSC | GC (%) |
|---|---|---|---|---|---|---|---|---|---|
| *Corydalis humilis* | ESK21-178 | 186,583 | 40.2 | 89,521 | 39.2 | 48,433 | 41.3 | 196 | 18.4 |
| *C. lineariloba* | ESK21-296 | 186,724 | 40.2 | 89,663 | 39.2 | 47,830 | 41.3 | 1,401 | 35.1 |
| *C. maculata* | ESK21-209 | 185,060 | 40.2 | 89,275 | 39.1 | 47,751 | 41.3 | 283 | 19.1 |
| *C. grandicalyx* | ESK21-300 | 189,792 | 40.3 | 91,225 | 39.4 | 49,132 | 41.2 | 303 | 21.1 |
| *C. hallaisanensis* | ESK21-125 | 186,162 | 40.2 | 89,553 | 39.2 | 48,185 | 41.3 | 239 | 17.6 |
| *C. remota* | ESK21-050 | 186,899 | 40.2 | 89,702 | 39.2 | 48,490 | 41.3 | 217 | 18.9 |
| *C. alata* | ESK21-020 | 190,434 | 40.2 | 89,854 | 39.2 | 50,156 | 41.3 | 268 | 16 |
| *C. misandra* | ESK21-176 | 186,851 | 40.2 | 89,637 | 39.1 | 48,441 | 41.2 | 332 | 15.7 |
| *C. namdoensis* | ESK21-018 | 184,889 | 40.2 | 87,743 | 39.2 | 48,419 | 41.2 | 308 | 15.6 |
| *C. bonghwaensis* | ESK22-006 | 188,863 | 40.2 | 89,397 | 39.1 | 49,573 | 41.3 | 320 | 17.8 |
| *C. cornupetala* | ESK21-064 | 184,406 | 40.3 | 89,979 | 39.2 | 47,099 | 41.4 | 229 | 15.7 |
| *C. filistipes* | CF1 | 189,459 | 40.4 | 92,730 | 39.5 | 48,279 | 41.4 | 171 | 21.1 |
| *C. turtschaninovii* | ESK21-189 | 195,713 | 40.4 | 89,940 | 39.1 | 52,755 | 41.4 | 263 | 19 |
| *C. wandoensis* | 2022_9 | 188,610 | 40.1 | 91,781 | 39.2 | 48,283 | 41.1 | 263 | 19 |
| *C. yanhusuo* | SRR15001456 | 187,922 | 40.1 | 89,400 | 39.2 | 49,152 | 41 | 218 | 21.6 |
| *C. intermedia* | ERR5554493 | 184,999 | 40.3 | 88,483 | 39.1 | 48,222 | 41.4 | 72 | 36.1 |
| *C. solida* | ERR5555065 | 185,904 | 40.2 | 89,397 | 39 | 48,138 | 41.3 | 231 | 31.2 |
| *C. ternata* | ESK22-077 | 187,819 | 41.1 | 95,294 | 39.9 | 38,840 | 43.4 | 14,845 | 36.9 |
| *C. hsiaowutaishanensis* | MT920561 | 188,784 | 40.8 | 88,558 | 39.4 | 44,070 | 42.8 | 12,086 | 37 |
| *C. ledebouriana* | KG_20180407_021 | 167,523 | 40.9 | 84,466 | 39.2 | 28,594 | 44 | 25,869 | 39.3 |
| *C. mucronifera* | SRR16375367 | 149,919 | 40.6 | 84,097 | 38.8 | 23,082 | 45.2 | 19,658 | 37.3 |
| *C. hendersonii* var. *altocristata* | SRR16668413 | 155,672 | 40.6 | 90,620 | 39 | 22,955 | 45.2 | 19,142 | 37.2 |
| *C. boweri* | SRR16685835 | 151,691 | 40.6 | 84,095 | 38.8 | 24,252 | 45 | 19,092 | 37.2 |
| *C. impatiens* | SRR16668463 | 175,380 | 40.8 | 89,674 | 39.5 | 42,515 | 42.3 | 676 | 33.7 |
| *C. conspersa* | MN843953 | 187,810 | 40.8 | 92,280 | 39.6 | 47,375 | 42.1 | 780 | 33.1 |
| *C. hendersonii* | SRR16668040 | 171,892 | 40.7 | 89,153 | 39.3 | 41,229 | 42.2 | 281 | 28.5 |
| *C. inopinata* | MT755641 | 181,335 | 40.9 | 91,727 | 39.9 | 44,053 | 42.2 | 1,502 | 34.4 |
| *C. trisecta* | MN654110 | 164,354 | 41.5 | 91,046 | 39.8 | 28,345 | 45.1 | 16,618 | 38.2 |
| *C. pauciovulata* | 2022_11 | 159,167 | 41.5 | 89,178 | 39.4 | 22,777 | 46.5 | 24,435 | 39.6 |
| *C. raddeana* | 2022_12 | 182,287 | 41.3 | 90,800 | 39.5 | 45,618 | 43.2 | 251 | 31.1 |
| *C. lupinoides* | SRR14352120 | 178,650 | 40.8 | 85,220 | 39.2 | 46,377 | 42.3 | 676 | 33.9 |
| *C. davidii* | MT920560 | 165,416 | 40.7 | 85,352 | 39.1 | 39,867 | 42.4 | 330 | 24.8 |
| *C. chrysosphaera* | MZ983401 | 200,923 | 40.7 | 90,710 | 39.1 | 54,930 | 42 | 353 | 29.5 |
| *C. capnoides* | SRR16761252 | 199,294 | 40.5 | 89,691 | 39.1 | 54,625 | 41.8 | 353 | 29.5 |
| *C. incisa* | ESK22-044 | 195,665 | 40.3 | 89,655 | 39.1 | 52,875 | 41.4 | 260 | 25.8 |
| *C. temulifolia* | MT920558 | 194,096 | 40.2 | 95,239 | 39.1 | 49,258 | 41.3 | 341 | 31.7 |
| *C. shensiana* | MW110634 | 155,935 | 40.6 | 82,752 | 38.9 | 26,344 | 44.7 | 20,495 | 36.8 |
| *C. bungeana* | ESK22-075 | 183,672 | 40.6 | 86,029 | 39.3 | 48,665 | 41.7 | 313 | 30.4 |
| *C. edulis* | MW110633 | 154,395 | 40.2 | 82,391 | 38.6 | 26,253 | 44.4 | 19,498 | 35.8 |
| *C. fangshanensis* | MZ440305 | 192,554 | 40.3 | 98,393 | 39.2 | 42,263 | 42.1 | 9,635 | 35.2 |
| *C. tomentella* | MT077878 | 190,196 | 40.2 | 96,531 | 39 | 42,000 | 42.2 | 9,665 | 35.4 |
| *C. saxicola* | MT920562 | 188,060 | 40.2 | 94,289 | 39 | 41,969 | 42.2 | 9,833 | 35.2 |
| *C. heterocarpa* | 2022_8 | 194,615 | 40.3 | 97,204 | 39.2 | 43,450 | 42.2 | 10,511 | 34.9 |
| *C. platycarpa* | ESK21-230 | 194,199 | 40.4 | 97,077 | 39.3 | 43,340 | 42.2 | 10,442 | 35 |
| *C. speciosa* | 2022_7 | 193,108 | 40.5 | 97,022 | 39.4 | 42,784 | 42.4 | 10,518 | 35 |
| *C. heracleifolia* | SRR15672852 | 190,029 | 40.6 | 90,328 | 38.8 | 43,920 | 43.2 | 11,861 | 34.9 |
| *C. semenowii* | KY-14-2 | 168,780 | 40.3 | 90,621 | 38.9 | 29,031 | 43.6 | 20,097 | 37.1 |

*(Continued)*

**Table 1.** (Continued)

| Scientific name | Accession | Total length | GC (%) | LSC | GC (%) | IRs | GC (%) | SSC | GC (%) |
|---|---|---|---|---|---|---|---|---|---|
| *C. adunca* | MT920559 | 196,118 | 41 | 92,145 | 39.8 | 47,221 | 42.8 | 9,531 | 34.7 |
| *Fumaria officinalis* | K004871 | 190,247 | 40.2 | 86,672 | 39 | 48,853 | 41.7 | 5,869 | 33.8 |
| *Lamprocapnos spectabilis* | MG873498 | 188,754 | 39.2 | 84,341 | 38.2 | 51,384 | 40.1 | 1,645 | 35 |
| *Papaver nudicaule* | MW151698 | 153,903 | 38.9 | 83,676 | 37.3 | 25,680 | 43.2 | 18,867 | 33.9 |
| *Stylophorum lasiocarpum* | MW232434 | 153,196 | 38.9 | 83,230 | 37.4 | 25,789 | 43.2 | 18,388 | 34 |
| *Meconopsis integrifolia* | MK533647 | 151,860 | 38.8 | 82,809 | 37.4 | 25,649 | 43 | 17,753 | 33.3 |
| *Chelidonium majus* | MK433200 | 159,741 | 38.7 | 87,697 | 37.2 | 26,735 | 43.1 | 18,574 | 33.3 |
| *Coreanomecon hylomeconoides* | KT274030 | 158,824 | 38.7 | 86,914 | 37.3 | 26,686 | 43.2 | 18,538 | 32.8 |
| *Macleaya cordata* | MT178411 | 163,179 | 38.6 | 87,933 | 37 | 28,329 | 42.8 | 18,588 | 33.1 |
| *Eschscholzia californica* | MK281585 | 160,201 | 38.7 | 88,147 | 37.2 | 26,781 | 43.1 | 18,492 | 33.1 |
| *Euptelea pleiosperma* | KU204900 | 161,834 | 38.6 | 90,449 | 37.1 | 26,037 | 43.3 | 19,311 | 33.3 |
| | AVERAGE | 182,552 | 40.2 | 89,496 | 38.9 | 40,847 | 42.5 | 7,953 | 30.2 |

GC, guanine-cytosine; LSC, large single-copy; IRs, inverted regions; SSC, small single-copy region; red highlights, maximum value; yellow highlights, minimum value.

In addition, the chloroplast genomes showed very diverse gene compositions. The number of CDSs ranged from 64 (*C. trisecta*) to 79 (*Meconopsis integrifolia*, *Chelidonium majus*, *Coreanomecon hylomeconoides*, *Macleaya cordata*, *Eschscholzia californica*, and *E. pleiosperma*), and the number of tRNA genes ranged from 28 (*C. ternata* and *C. trisecta*) to 34 (*C. temulifolia* and *C. tomentella*). However, all genomes contained exactly four rRNAs. The total number of genes ranged from 96 (*C. trisecta*) to 115 (*C. temulifolia*), and the number of pseudogenes was 9 in most cases (*C. inopinata* and *C. trisecta*). In the LSC region, *C. trisecta* had the least number of genes (55), whereas *Papaver nudicaule*, *Stylophorum lasiocarpum*, *Meconopsis integrifolia*, *Chelidonium majus*, *Coreanomecon hylomeconoides*, *Eschscholzia californica*, and *E. pleiosperma* had the most (61). The number of tRNA genes was the lowest (20) in *C. trisecta* and *L. spectabilis* and was the highest (26) in *C. temulifolia* and *C. tomentella*. In the SSC region, the number of genes ranged from 0 to 12 (*C. shensiana*, *Meconopsis integrifolia*, *Chelidonium majus*, *Coreanomecon hylomeconoides*, *Macleaya cordata*, *Eschscholzia californica*, and *E. pleiosperma*). There were no (or only one) tRNA genes, and five pseudogenes were the most common. In IRs, the number of genes ranged from 4 (*C. mucronifera*, *C. hendersonii* var. *altocristata*, *C. boweri*, *C. trisecta*, *C. pauciovulata*) to 18 (*L. spectabilis*). The number of tRNAs ranged from 6 (*C. trisecta*) to 12 (*L. spectabilis*; Table 2). A total of 17 genes contained one intron, and the *pafI* gene contained two introns. The remaining nine CDSs (*atpF*, *ndhA*, *ndhB*, *petB*, *petD*, *rpl16*, *rpl2*, *rpoC1*, and *rps16*) and six tRNA genes (*trnA-UGC*, *trnG-GCC*, *trnI-GAU*, *trnK-UUU*, *trnL-UAA*, and *trnV-UAC*) contained one intron. The *rps12* gene was confirmed to be a trans-spliced gene consisting of three exons: exon 1 in the LSC region, and exons 2 and 3 in the IR regions. However, the introns of *rpl16* and *rps16* were deleted in *C. ternata* (S1 Fig). Furthermore, the *accD* gene was lost in *Corydalis*, *Fumaria*, and *Lamprocapnos*.

### 3.2. Types, structures, and evolutionary patterns of chloroplast genomes in the Papaveraceae

A total of 58 chloroplast genomes—including 36 newly assembled chloroplast genomes—were compared. We identified nine types of rearrangement events in the chloroplast genome structures of *Corydalis* spp. (excluding the *L. spectabilis* type; Fig 2). The nine types of complete chloroplast genomes are described in detail below.

**Table 2. Genes identified in the chloroplast genomes of 57 species of Papaveraceae and one species of Eupteleaceae (outgroup).**

| Scientific name | Total genes | Total | | | | LSC | | | SSC | | | IRs | | | |
|---|---|---|---|---|---|---|---|---|---|---|---|---|---|---|---|
| | | CDS | tRNA | rRNA | ψ | CDS | tRNA | ψ | CDS | tRNA | ψ | CDS | tRNA | rRNA | ψ |
| *Corydalis humilis* | 112 | 78 | 30 | 4 | 1 | 61 | 22 | 1 | 0 | 0 | 0 | 17 | 8 | 4 | 0 |
| *C. lineariloba* | 112 | 78 | 30 | 4 | 1 | 61 | 22 | 1 | 0 | 0 | 0 | 17 | 8 | 4 | 0 |
| *C. maculata* | 112 | 78 | 30 | 4 | 1 | 61 | 22 | 1 | 0 | 0 | 0 | 17 | 8 | 4 | 0 |
| *C. grandicalyx* | 112 | 78 | 30 | 4 | 1 | 61 | 22 | 1 | 0 | 0 | 0 | 17 | 8 | 4 | 0 |
| *C. hallaisanensis* | 112 | 78 | 30 | 4 | 1 | 61 | 22 | 1 | 0 | 0 | 0 | 17 | 8 | 4 | 0 |
| *C. remota* | 112 | 78 | 30 | 4 | 1 | 61 | 22 | 1 | 0 | 0 | 0 | 17 | 8 | 4 | 0 |
| *C. alata* | 112 | 78 | 30 | 4 | 1 | 61 | 22 | 1 | 0 | 0 | 0 | 17 | 8 | 4 | 0 |
| *C. misandra* | 112 | 78 | 30 | 4 | 1 | 61 | 22 | 1 | 0 | 0 | 0 | 17 | 8 | 4 | 0 |
| *C. namdoensis* | 112 | 78 | 30 | 4 | 1 | 61 | 22 | 1 | 0 | 0 | 0 | 17 | 8 | 4 | 0 |
| *C. bonghwaensis* | 112 | 78 | 30 | 4 | 1 | 61 | 22 | 1 | 0 | 0 | 0 | 17 | 8 | 4 | 0 |
| *C. cornupetala* | 112 | 78 | 30 | 4 | 1 | 61 | 22 | 1 | 0 | 0 | 0 | 17 | 8 | 4 | 0 |
| *C. filistipes* | 112 | 78 | 30 | 4 | 1 | 61 | 22 | 1 | 0 | 0 | 0 | 17 | 8 | 4 | 0 |
| *C. turtschaninovii* | 112 | 78 | 30 | 4 | 1 | 61 | 22 | 1 | 0 | 0 | 0 | 17 | 8 | 4 | 0 |
| *C. wandoensis* | 112 | 78 | 30 | 4 | 1 | 61 | 22 | 1 | 0 | 0 | 0 | 17 | 8 | 4 | 0 |
| *C. yanhusuo* | 112 | 78 | 30 | 4 | 1 | 61 | 22 | 1 | 0 | 0 | 0 | 17 | 8 | 4 | 0 |
| *C. intermedia* | 112 | 78 | 30 | 4 | 1 | 61 | 22 | 1 | 0 | 0 | 0 | 17 | 8 | 4 | 0 |
| *C. solida* | 112 | 78 | 30 | 4 | 1 | 61 | 22 | 1 | 0 | 0 | 0 | 17 | 8 | 4 | 0 |
| *C. ternata* | 110 | 78 | 28 | 4 | 2 | 61 | 20 | 2 | 6 | 0 | 0 | 11 | 8 | 4 | 0 |
| *C. hsiaowutaishanensis* | 112 | 78 | 30 | 4 | 1 | 61 | 22 | 1 | 4 | 0 | 0 | 13 | 8 | 4 | 0 |
| *C. ledebouriana* | 105 | 71 | 30 | 4 | 3 | 58 | 22 | 1 | 8 | 1 | 2 | 5 | 7 | 4 | 0 |
| *C. mucronifera* | 102 | 68 | 30 | 4 | 6 | 58 | 22 | 1 | 6 | 1 | 5 | 4 | 7 | 4 | 0 |
| *C. hendersonii* var. *altocristata* | 101 | 67 | 30 | 4 | 7 | 57 | 22 | 2 | 6 | 1 | 5 | 4 | 7 | 4 | 0 |
| *C. boweri* | 103 | 69 | 30 | 4 | 5 | 58 | 22 | 1 | 7 | 1 | 4 | 4 | 7 | 4 | 0 |
| *C. impatiens* | 104 | 70 | 30 | 4 | 7 | 58 | 22 | 4 | 0 | 0 | 0 | 12 | 8 | 4 | 3 |
| *C. conspersa* | 102 | 68 | 30 | 4 | 4 | 58 | 22 | 1 | 0 | 0 | 0 | 10 | 8 | 4 | 3 |
| *C. hendersonii* | 103 | 69 | 30 | 4 | 3 | 58 | 22 | 1 | 0 | 0 | 0 | 11 | 8 | 4 | 2 |
| *C. inopinata* | 101 | 68 | 29 | 4 | 9 | 58 | 21 | 4 | 1 | 0 | 1 | 9 | 8 | 4 | 4 |
| *C. trisecta* | 96 | 64 | 28 | 4 | 9 | 55 | 21 | 3 | 5 | 1 | 4 | 4 | 6 | 4 | 2 |
| *C. pauciovulata* | 101 | 68 | 29 | 4 | 7 | 58 | 21 | 3 | 6 | 1 | 3 | 4 | 7 | 4 | 1 |
| *C. raddeana* | 101 | 68 | 29 | 4 | 8 | 58 | 21 | 4 | 0 | 0 | 0 | 10 | 8 | 4 | 4 |
| *C. lupinoides* | 101 | 68 | 29 | 4 | 3 | 57 | 21 | 1 | 0 | 0 | 0 | 11 | 8 | 4 | 2 |
| *C. davidii* | 101 | 67 | 30 | 4 | 3 | 58 | 22 | 1 | 0 | 0 | 0 | 9 | 8 | 4 | 2 |
| *C. chrysosphaera* | 112 | 78 | 30 | 4 | 1 | 61 | 22 | 1 | 0 | 0 | 0 | 17 | 8 | 4 | 0 |
| *C. capnoides* | 112 | 78 | 30 | 4 | 1 | 61 | 22 | 1 | 0 | 0 | 0 | 17 | 8 | 4 | 0 |
| *C. incisa* | 110 | 77 | 29 | 4 | 3 | 61 | 21 | 2 | 0 | 0 | 0 | 16 | 8 | 4 | 1 |
| *C. temulifolia* | 116 | 78 | 34 | 4 | 1 | 61 | 26 | 1 | 0 | 0 | 0 | 17 | 8 | 4 | 0 |
| *C. shensiana* | 110 | 77 | 29 | 4 | 3 | 60 | 21 | 2 | 12 | 1 | 0 | 5 | 7 | 4 | 1 |
| *C. bungeana* | 104 | 71 | 29 | 4 | 3 | 59 | 21 | 1 | 0 | 0 | 0 | 12 | 8 | 4 | 2 |
| *C. edulis* | 108 | 75 | 29 | 4 | 2 | 59 | 21 | 1 | 11 | 1 | 0 | 5 | 7 | 4 | 1 |
| *C. fangshanensis* | 111 | 77 | 30 | 4 | 3 | 60 | 22 | 2 | 3 | 0 | 1 | 14 | 8 | 4 | 0 |
| *C. tomentella* | 115 | 77 | 34 | 4 | 2 | 60 | 26 | 1 | 4 | 0 | 1 | 13 | 8 | 4 | 0 |
| *C. saxicola* | 111 | 77 | 30 | 4 | 3 | 60 | 22 | 2 | 4 | 0 | 1 | 13 | 8 | 4 | 0 |
| *C. heterocarpa* | 112 | 78 | 30 | 4 | 2 | 61 | 22 | 1 | 4 | 0 | 1 | 13 | 8 | 4 | 0 |
| *C. platycarpa* | 112 | 78 | 30 | 4 | 2 | 61 | 22 | 1 | 4 | 0 | 1 | 13 | 8 | 4 | 0 |
| *C. speciosa* | 111 | 77 | 30 | 4 | 2 | 60 | 22 | 2 | 4 | 0 | 0 | 13 | 8 | 4 | 0 |
| *C. heracleifolia* | 111 | 77 | 30 | 4 | 3 | 60 | 21 | 3 | 5 | 0 | 0 | 12 | 9 | 4 | 0 |

*(Continued)*

**Table 2.** (Continued)

| Scientific name | Total genes | Total | | | | LSC | | | SSC | | | IRs | | | |
|---|---|---|---|---|---|---|---|---|---|---|---|---|---|---|---|
| | | CDS | tRNA | rRNA | ψ | CDS | tRNA | ψ | CDS | tRNA | ψ | CDS | tRNA | rRNA | ψ |
| *C. semenowii* | 112 | 78 | 30 | 4 | 2 | 61 | 22 | 2 | 11 | 1 | 0 | 6 | 7 | 4 | 0 |
| *C. adunca* | 106 | 72 | 30 | 4 | 5 | 59 | 22 | 2 | 2 | 0 | 1 | 11 | 8 | 4 | 2 |
| *Fumaria officinalis* | 113 | 78 | 31 | 4 | 1 | 60 | 22 | 1 | 1 | 0 | 0 | 17 | 9 | 4 | 0 |
| *Lamprocapnos spectabilis* | 115 | 79 | 32 | 4 | 1 | 60 | 20 | 1 | 1 | 0 | 0 | 18 | 12 | 4 | 0 |
| *Papaver nudicaule* | 113 | 78 | 31 | 4 | 0 | 61 | 23 | 0 | 11 | 1 | 0 | 6 | 7 | 4 | 0 |
| *Stylophorum lasiocarpum* | 112 | 78 | 30 | 4 | 1 | 61 | 22 | 0 | 11 | 1 | 1 | 6 | 7 | 4 | 0 |
| *Meconopsis integrifolia* | 113 | 79 | 30 | 4 | 0 | 61 | 22 | 0 | 12 | 1 | 0 | 6 | 7 | 4 | 0 |
| *Chelidonium majus* | 113 | 79 | 30 | 4 | 0 | 61 | 22 | 0 | 12 | 1 | 0 | 6 | 7 | 4 | 0 |
| *Coreanomecon hylomeconoides* | 113 | 79 | 30 | 4 | 0 | 61 | 22 | 0 | 12 | 1 | 0 | 6 | 7 | 4 | 0 |
| *Macleaya cordata* | 113 | 79 | 30 | 4 | 0 | 58 | 22 | 0 | 12 | 1 | 0 | 9 | 7 | 4 | 0 |
| *Eschscholzia californica* | 113 | 79 | 30 | 4 | 0 | 61 | 22 | 0 | 12 | 1 | 0 | 6 | 7 | 4 | 0 |
| *Euptelea pleiosperma* | 113 | 79 | 30 | 4 | 0 | 61 | 22 | 0 | 12 | 1 | 0 | 6 | 7 | 4 | 0 |
| **Average** | 109 | 75 | 30 | 4 | 3 | 60 | 22 | 1 | 4 | 0 | 1 | 12 | 8 | 4 | 1 |

CDS, protein coding genes; ψ, Pseudogene; LSC, large single-copy; IRs, inverted regions; SSC, small single-copy region; red highlights, maximum value; yellow highlights, minimum values.

**3.2.1. The complete chloroplast genome structure of *Fumaria officinalis*.** Compared with *E. pleiosperm* (outgroup), a 42,378 bp region of the genome (ranging from *trnT-UGU* to *trnQ-UUG*) was found to be reversed in the LSC of *F. officinalis*. In addition, an IR reversed a previously reported 13,938 bp region (ranging from *trnR-ACG* to *ndhB*) in *Corydalis* spp. Specifically, the region from *rps16* to *trnT-UGU* moved between *ndhF* and *rpl32* and was incorporated into the IR region through IR extension. Three *trnT-UGU* genes were identified (S2 Fig).

**3.2.2. The complete chloroplast genome structure of Type 1 in *Corydalis*.** Type 1 had been previously reported in a study on two *Corydalis* spp. (*C. edulis* and *C. shensiana*) [16]. The genome of Type 1 plants showed a typical quaternary structure known in angiosperms. The *accD* gene was lost in Type 1, and the *ycf1* gene was pseudogenized in *C. edulis* (S3 Fig).

**3.2.3. The complete chloroplast genome structure of Type 2 in *Corydalis*.** Type 2 of *Corydalis* was identified in *C. adunca*, *C. semenowii*, *C. heracleifolia*, and *C. trisecta*. As seen in *F. officinalis*, the region from *trnR-ACG* to *ndhB* was reversed in the IR region. Moreover, the *ndhF* gene was incorporated into the IR region (extended to *ndhG* in *C. heracleifolia*; extended to part of the *ndhA* in *C. adunca*). Unlike in other *Corydalis* species, the *accD* gene was not lost and remained as a pseudogene in *C. semenowii* and *C. heracleifolia*. The *ndhA*, *ndhC*, *ndhD*, *ndhF*, and *ndhH* genes were pseudogenized in *C. adunca*, and *ndhI* was completely lost. As in *F. officinalis*, the *rps16* gene moved between *ndhF* and *rpl32*. The *matK* gene was identified as a pseudogene only in *C. trisecta* (S4 Fig) [39].

**3.2.4. The complete chloroplast genome structure of Type 3 in *Corydalis*.** Type 3 of *Corydalis* was identified in *C. speciosa*, *C. platycarpa*, *C. heterocarpa*, *C. tomentella*, *C. saxicola*, and *C. fangshanensis*. Compared with Type 2, Type 3 showed an inversion in the region from *rbcL* to *trnV-UAC*, and it shifted between *atpH* and *atpI*. In all species belonging to Type 3, the IR was extended to a part of the exon 2 of *ndhA* (S5 Fig). In three species (*C. speciosa*, *C. platycarpa*, and *C. heterocarpa*), the position of the *rpl23* gene had shifted from between *rpl2* and *ycf2* to between *ndhF* and *rpl32*.

**3.2.5. The complete chloroplast genome structure Type 4 in *Corydalis*.** Type 4 of *Corydalis* was identified in *C. boweri*, *C. mucronifera*, *C. hendersonii* var. *altocristata*, *C. ledebouriana*,

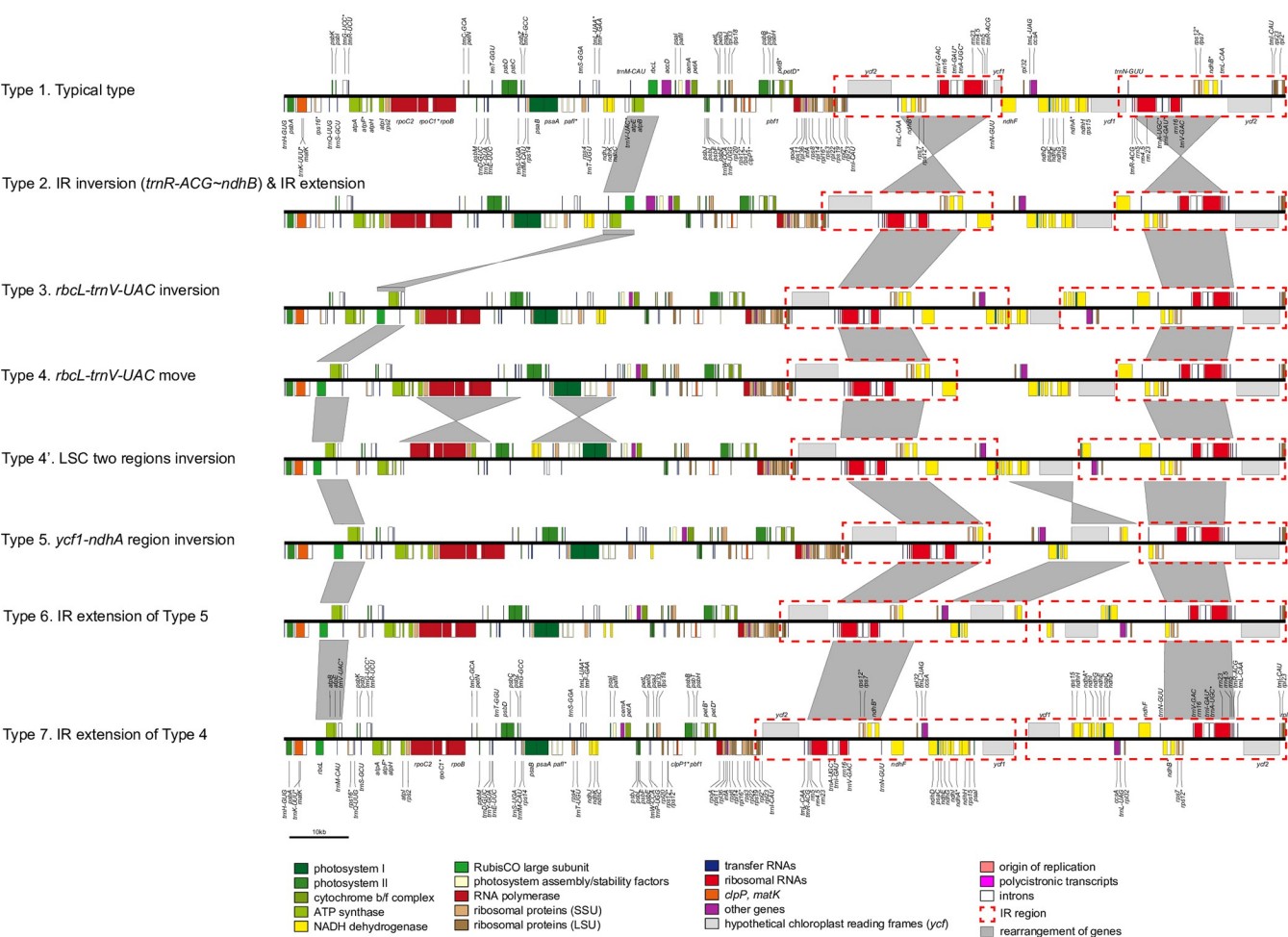

**Fig 2. Types of chloroplast genomes in *Corydalis* species.** The red dotted boxes represent inverted repeats, and the gray boxes represent reversal and relocation events.

and *C. hsiaowutaishanensis*. Compared with Type 3, the region from *rbcL* to *trnV-UAC* shifted from between *atpH* and *atpI* to between *matK* and *rps16* in Type 4. The IR extended to *ndhF* in general, but extended to a part of the *ndhI* gene in *C. hsiaowutaishanensis*. The regions containing *ndhJ*, *ndhK*, and *ndhC* had been deleted in all species except *C. hsiaowutaishanensis*. In addition, the pseudogenes of the *ndh* genes differed between species (S6 Fig).

**3.2.6. The complete chloroplast genome structure Type 4' in *Corydalis*.** Type 4' of *Corydalis* was identified in only one species (*C. ternata*). Compared with Type 4, Type 4' genomes showed three major differences. First, a 14,406 bp region (including *atpI* and *petN*) was inverted. Second, a 20,057 bp region (including *trnfM-CAU* and *ndhC*) was also inverted. Finally, the introns of the *rps16* and *rpl16* genes were lost. The IR extended to a portion of the *ndhE* gene (S7 Fig).

**3.2.7. The complete chloroplast genome structure Type 5 in *Corydalis*.** Type 5 of *Corydalis* was identified in only one species (*C. pauciovulata*). Compared with Type 4, Type 5 showed one big difference in that a 12,852 bp region between *ndhA* and *ycf1* in the SSC region was inverted. Among the *ndh* genes, *ndhA*, *ndhC*, *ndhF*, *ndhI*, and *ndhK* were lost; *ndhB*, *ndhD*, *ndhE*, *ndhG*, and *ndhJ* were identified as pseudogenes; and only the *ndhH* gene was present (S8 Fig).

**3.2.8. The complete chloroplast genome structure Type 6 in *Corydalis*.** Type 6 of *Corydalis* was identified in *C. davidii*, *C. lupinoides*, *C. raddeana*, *C. inopinata*, *C. hendersonii*, and *C. conspersa*. Type 6 showed an extended IR compared with Type 5, and the SSC region was shortened from 251 bp to 1,502 bp. There were no specific or consistent patterns of loss or pseudogenization of *ndh* genes among species (S9 Fig).

**3.2.9. The complete chloroplast genome structure Type 7 in *Corydalis*.** Type 7 of *Corydalis* was identified in 23 *Corydalis* spp.: *C. bungeana*, *C. temulifolia*, *C. incisa*, *C. chrysosphaera*, *C. capnoides*, *C. impatiens*, *C. solida*, *C. intermedia*, *C. yanhusuo*, *C. wandoensis*, *C. turtschaninovii*, *C. filistipes*, *C. cornupetala*, *C. bonghwaensis*, *C. namdoensis*, *C. misandra*, *C. alata*, *C. remota*, *C. hallaisanensis*, *C. grandicalyx*, *C. maculate*, *C. ohii*, and *C. humilis*. Thus, most of the analyzed *Corydalis* species had this Type of structure. Type 7 showed an extended IR compared with Type 4. The SSC region of Type 7 was shorter than those of all other types, lengths ranging from 72 bp to 1,401 bp, and contained no genes (S10 Fig).

### 3.3. Relative synonymous codon usage

The relative synonymous codon usage (RSCU) was calculated from the complete chloroplast genome sequences of 48 *Corydalis* spp. using all CDSs. The number of codons ranged from 13,529 (*C. trisecta*) to 22,425 (*C. ternata*; S4 Table). The most abundant amino acid was leucine (Leu; 10.62%), and the least abundant was cysteine (Cys; 1.09%). The most used codon was AAA [39,017; encodes lysine (Lys)], and the least used codon was UGC [2,924; encodes cysteine (Cys)]. RSCU frequency analysis revealed a bias in codon usage. Overall, 30 amino acids had an RSCU > 1, and two amino acids—methionine (AUG) and tryptophan (UGG)—showed no codon usage bias (RSCU = 1.00). The highest RSCU value was recorded for AGA [1.69; encodes arginine (Arg)], and the lowest values (0.4) were recorded for CGC and AGC [encode arginine (Arg) and serine (Ser), respectively] (Fig 3).

### 3.4. Simple sequence repeats (SSRs) and long repeat sequences

A total of 3,255 SSRs were identified in the chloroplast genomes of 48 *Corydalis* spp. and one *Fumaria* species (S5 and S6 Tables). The number of SSRs ranged from 34 (*C. edulis*) to 94 (*C.*

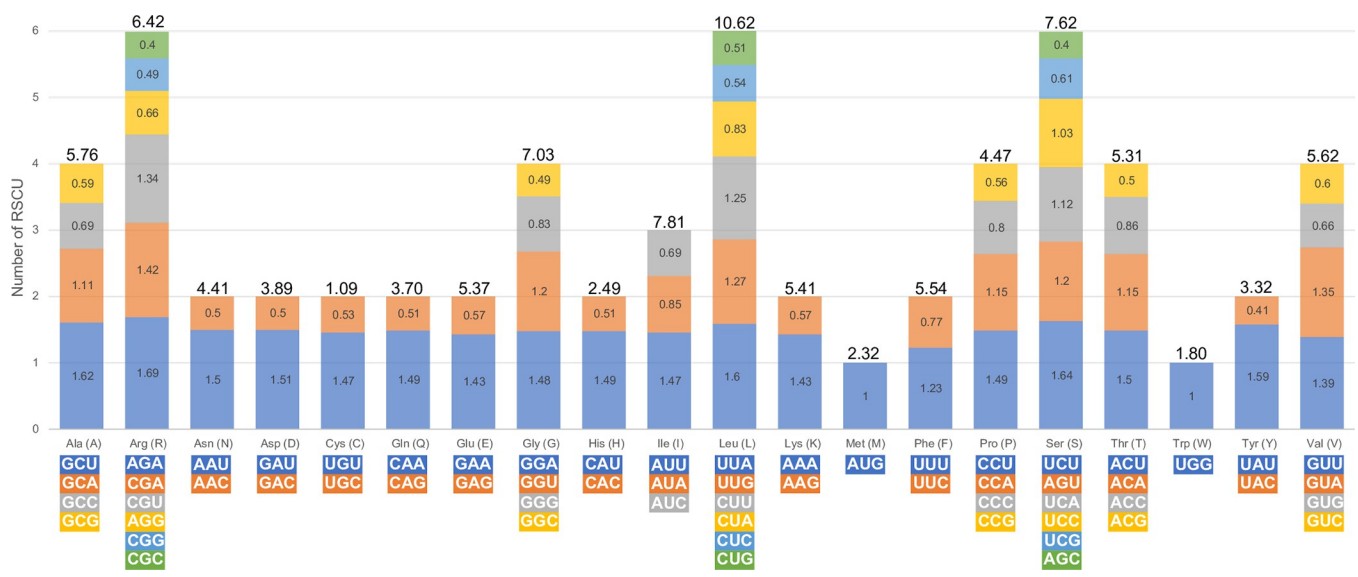

**Fig 3. Relative synonymous codon usage (RSCU) analysis of 20 amino acids in all coding sequences of the complete chloroplast genome of 48 *Corydalis* species.**

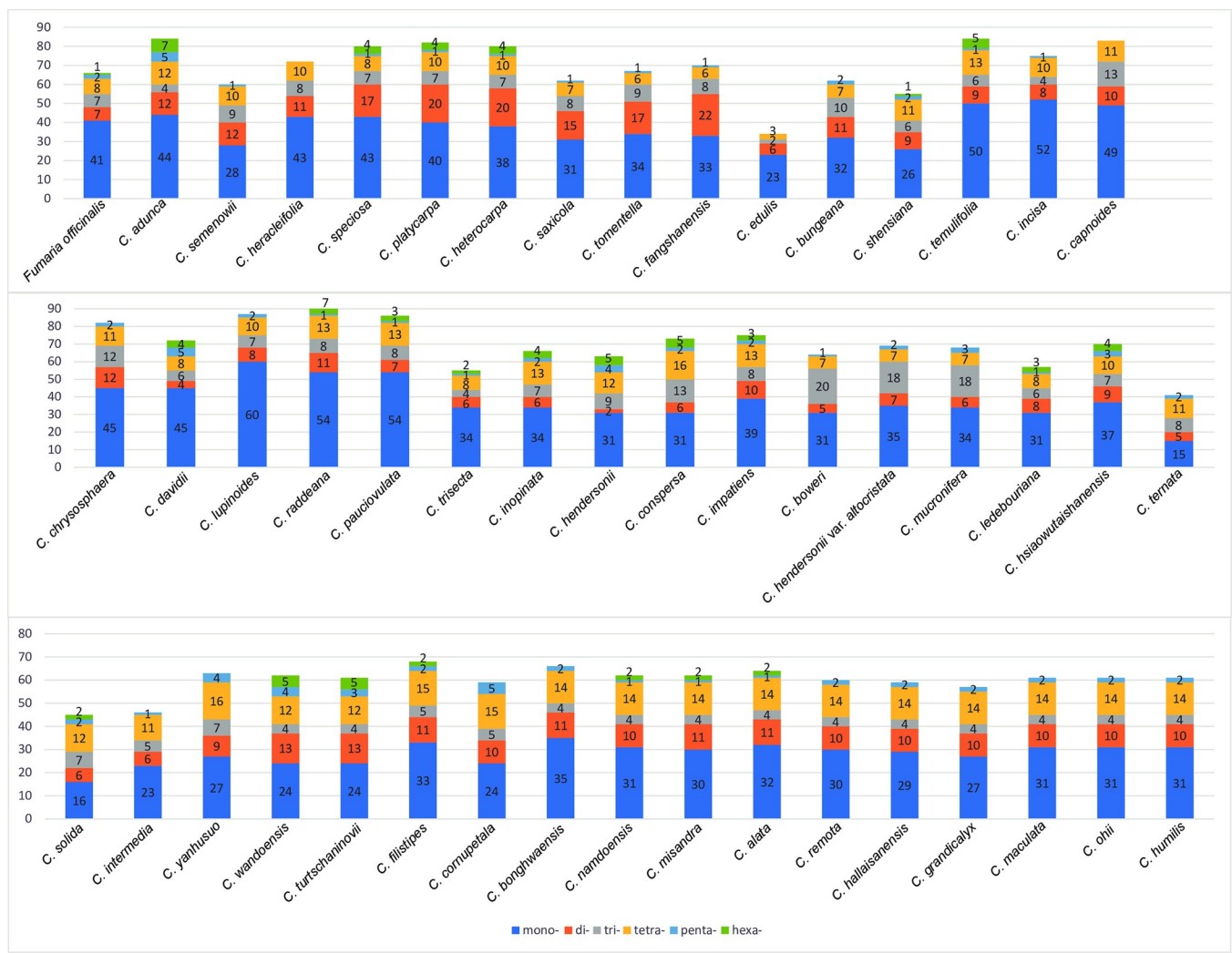

**Fig 4. The proportion of simple sequence repeats in the chloroplast genome of 48 *Corydalis* species and *Fumaria officinalis*.**

*raddeana*), and the average was 66.4 (S5 Table). Among the six types of SSRs (mono- to hexanucleotide), mononucleotide SSRs were the most common (1,695), accounting for 52.1% of all SSRs. Tetranucleotide and dinucleotide SSRs comprised 16.7% and 15% of all SSRs, respectively (Fig 4). Pentanucleotide SSRs were the least common (60; 1.8%). A/T repeats were the most common among mononucleotide SSRs (95.7%), AT/TA among dinucleotide SSRs (74%), ATC/TAG among trinucleotide SSRs (51.9%), AGAT/ATCT among tetranucleotide SSRs (29.2%), AAAAG/CTTTT and AATAG/ATTCT among pentanucleotide SSRs (20%), and AGCGAT/ATCGCT among hexanucleotide SSRs (35.6%; S6 Table).

We also examined long and complex repeat sequences that play an important role in determining the genome structure [60]. A total of 2,401 long repeat sequences were identified from the 49 chloroplast genomes (including 48 *Corydalis* spp. and one *Fumaria*; S7 Table). Reverse or complementary repeats were not found in any of the chloroplast genomes, and all repeats were determined to be forward and palindromic repeats. The number of forward repeats was the lowest (21) in *C. bungeana* and *C. wandoensis* and was the highest in *C. fangshanensis* (44). The number of palindromic repeats was the lowest in *C. fangshanensis* (5) and was the highest

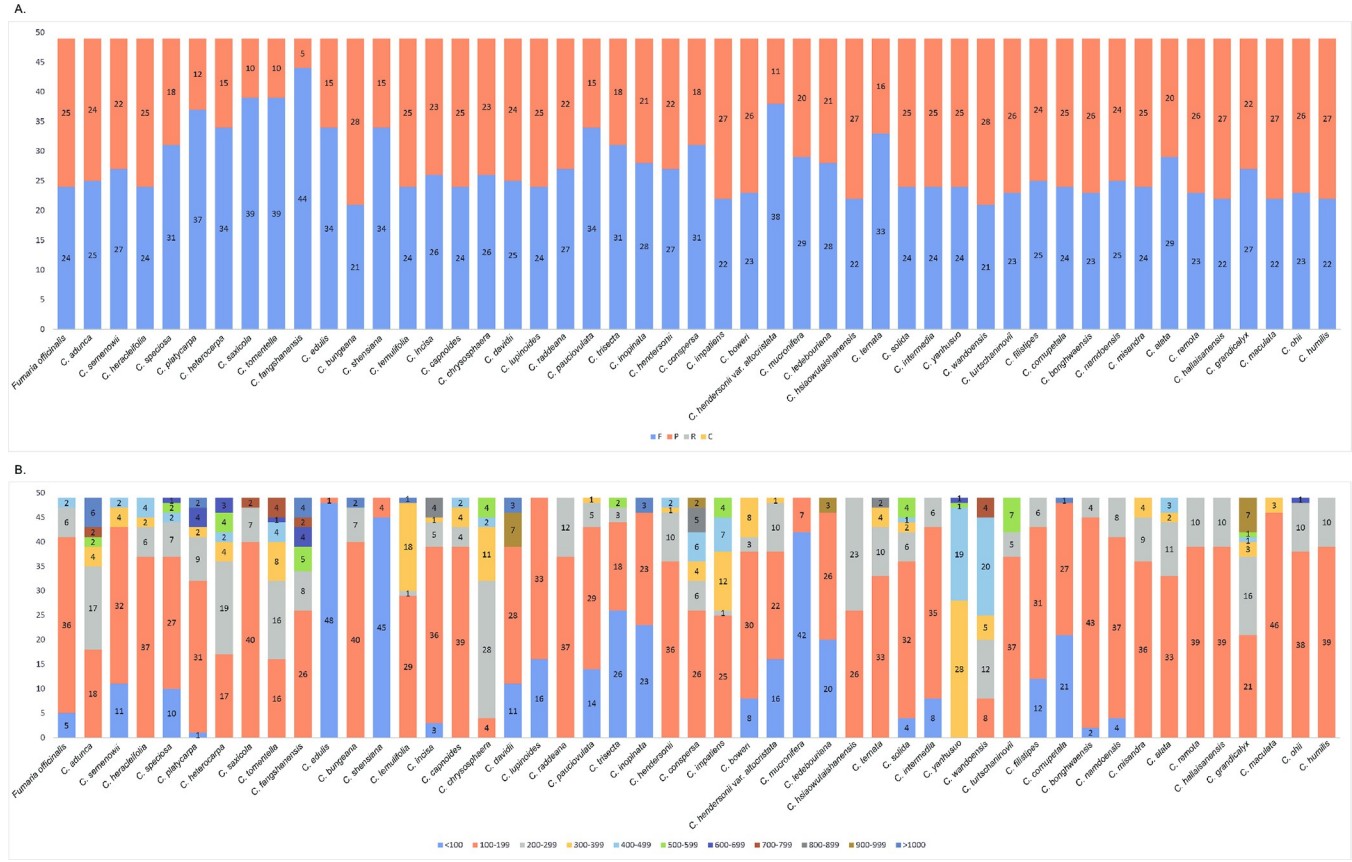

**Fig 5. Analysis of long repeated sequences in the chloroplast genomes of 48 *Corydalis* species and *Fumaria officinalis*.** (A) The number of four long repeat types. (B) Number of long repeat types by length.

(28) in *C. bungeana* and *C. wandoensis* (Fig 5A and S7 Table). The size of long and complex repeats ranged from 100 bp to 199 bp. Repeats with a length of ≥ 1,000 bp were identified in 9 of 49 species and were the most common in *C. adunca* (6; Fig 5B).

## 3.5. Comparison of the chloroplast genomes

Using the annotated *E. pleiosperma* chloroplast genome as a reference, we compared our generated sequences with those of taxa representing the nine types of chloroplast genomes. The LAGAN method (Fig 6A) revealed that the LSC of *Fumaria officinalis* had large unsortable regions from *rps16* to *trnT-UGU*. Type 4' of *Corydalis* contained a large region where two regions—from *atpI* to *petN* and from *trnfM-CAU* to *ndhC*—could not be aligned. In types 3–7 of *Corydalis*, alignment was not possible because of inversions and translocations in and between *trnV-UAC* and *rbcL*. Apart from Type 1 of *Corydalis*, all other types contained unalignable regions in the IR region from *ndhB* to *trnR-ACG*. In types 5 and 6 of *Corydalis*, alignment from *ndhI* to *ycf1* was not possible due to gene inversions. Conversely, all sequences were well-aligned using the Shuffle-LAGAN method (Fig 6B). Because there were many rearranged regions for each type of chloroplast genome, at least 14 newly aligned regions were identified in this study. Genes were generally more conserved than non-coding regions; in particular, the rRNA and tRNA regions of the IR region were found to be more conserved. However, we also found some *ndh* genes and some highly variable regions (such as *trnV-UAC*).

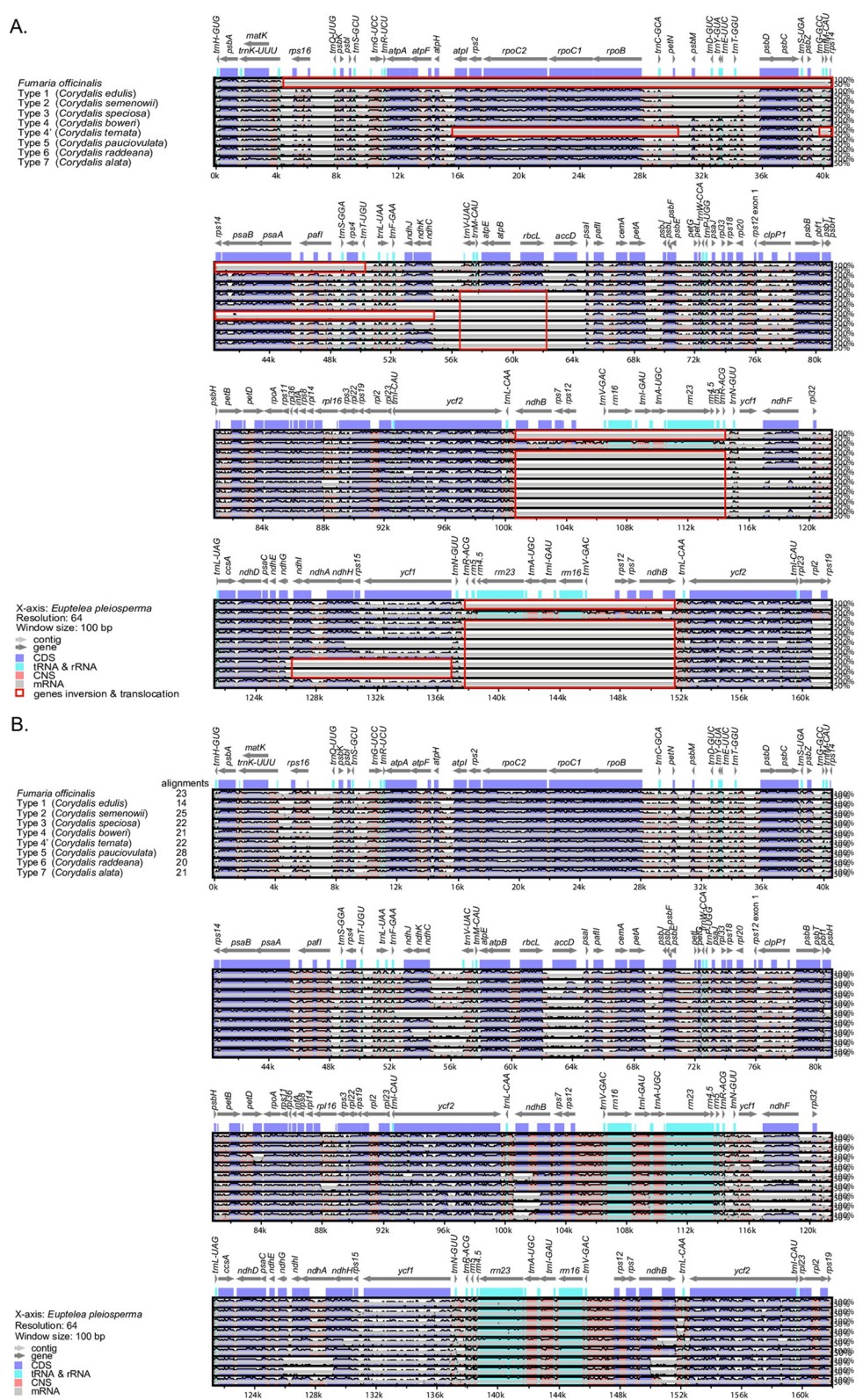

**Fig 6. Sequence identity plots of nine types of chloroplast genomes in *Corydalis* species.** *Euptelea pleiosperma* sequences have been used as references. (A) Using the LAGAN alignment. (B) Using the Shuffle-LAGAN alignment. Grey arrows indicate the orientation of genes, red bars represent non-coding sequences, purple bars represent exons, and blue bars represent introns; the vertical scale indicates the percentage identity within 50–100%.

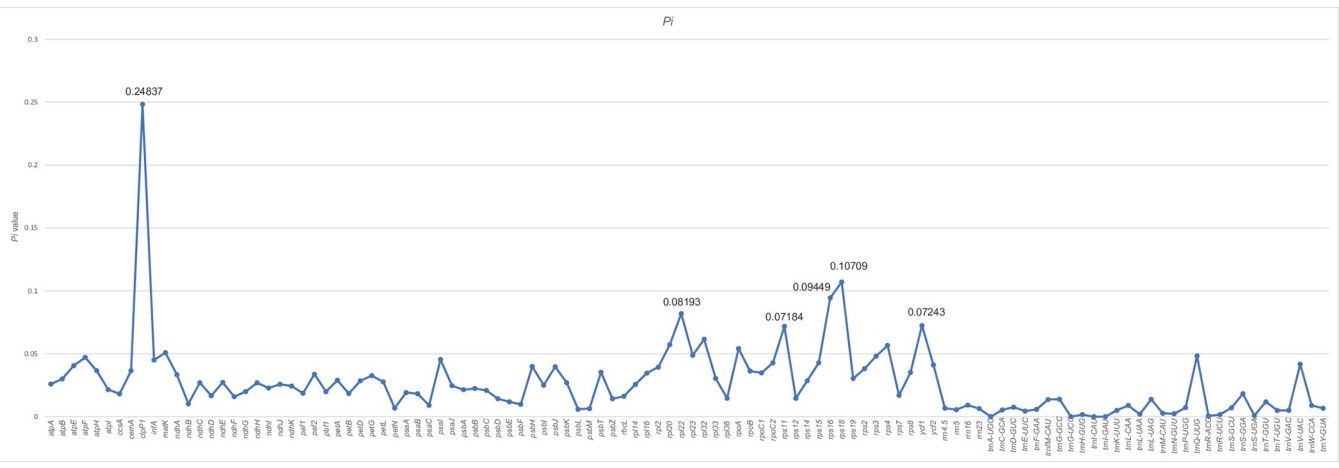

**Fig 7. Sliding window analysis of the nucleotide diversity (Pi) of genes in 48 *Corydalis* samples.**

### 3.6. Identification of divergence hotspots

To understand DNA polymorphisms (*Pi*), we calculated the nucleotide variability of 112 CDSs among the chloroplast genomes of 48 *Corydalis* species (Fig 7). These hotspot regions can be developed as molecular markers that could be useful for phylogenetic analysis. In addition, these regions can be used to develop DNA barcodes that can facilitate species identification in *Corydalis*. The *Pi* values in the coding domain ranged from 0 to 0.24837 (*clpP1*), with an average value of 0.027084. Overall, *clpP1* showed the highest *Pi* value (0.24837); its intron showed a *Pi* value of 0.09274 (data not shown), which was 2.68 times higher than that of the intergenic spacer.

### 3.7. Phylogenetic analyses

We used sequences from 58 species—57 species of Papaveraceae (including 36 newly assembled chloroplast genomes) and one species of Eupteleaceae as an outgroup—to infer ML-based and BI-based phylogenies. Both phylogenies (ML and BI) had the same topology and showed high support for each branch. Within Papaveraceae, two subfamilies—Papaveroideae and Eschscholzioideae—branched first, whereas the third subfamily (Fumarioideae) formed a monophyletic clade. Within *Corydalis*, the three subgenera (*Cremnocapnos*, *Sophorocapnos*, and *Corydalis*) were strongly supported as monophyletic groups. Subgenus *Cremnocapnos* diverged first, followed by subgenus *Sophorocapnos*, followed by two branches within subgenus *Corydalis* (Fig 8).

## 4. Discussion

### 4.1. The complete chloroplast genome of Fumaria officinalis

In this study, we generated the first complete chloroplast genome of *Fumaria officinalis* (Papaveraceae: Fumarioideae). To date, only two genera—*Corydalis* and *Lamprocapnos*—have been identified within Fumarioideae. Thus, we have newly added the genus *Fumaria* to this subfamily. The length of the chloroplast genome, LSC, SSC, and IR was 190,247 bp, 86,672 bp, 5,869 bp, and 48,853 bp, respectively; the GC content was 40.2%. The size of the genome expanded with the expansion of Irs. The gene contents included 78 CDSs, 31 tRNAs, and 4 rRNAs, and the *accD* gene was lost. Most of the complete chloroplast genomes in subfamily Fumarioideae showed signatures of rearrangement, especially *L. spectabilis*, which experienced at least six IR

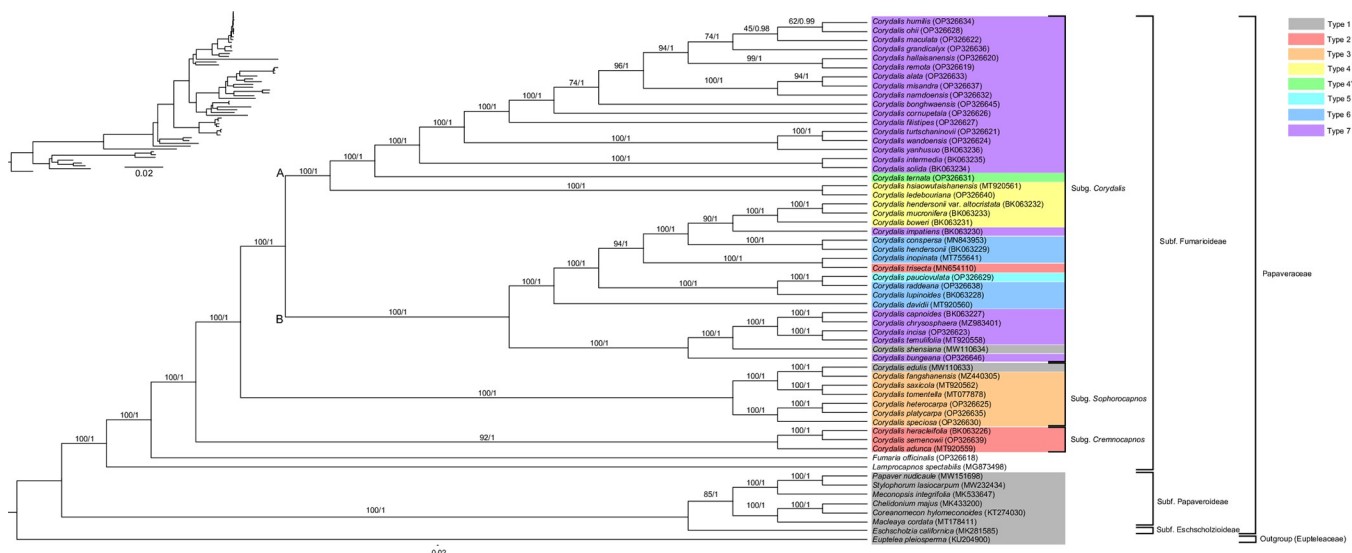

**Fig 8. Maximum likelihood (ML) and Bayesian inference (BI) phylogenetic trees based on 78 protein-coding genes from 58 Papaveraceae species.**
Bootstrap support and posterior probability (PP) values are shown at the nodes.

boundary shifts and five inversions [18]. As shown here, *F. officinalis* underwent two inversion processes and one relocation. The *trnT-UGU* gene is usually located between *rps4* and *trnL-UAA* [61]. However, in *F. officinalis*, this gene was rearranged along with the *rps16* gene. Therefore, we speculate that following the reversal in the LSC region, the gene was rearranged into the space between *ndhF* and *rpl32*. This structure has also been confirmed in *C. adunca* [23]. The SSR found in this study had higher A/T content and lower G/C content. In particular, the identified mononucleotide repeat showed an A/T content of 92.9%. Thus, the SSR identified here can be utilized as a molecular marker for population genetic studies in *Fumaria* spp. A total of 49 long repeat sequences were identified, and this number is identical to that reported for other species in subfamily Fumarioideae. Most repeats were in the 100–199 bp range, with the largest repeat being 487 bp in length. This may be further evidence of rearrangements in the chloroplast genome of *F. officinalis*. Thus, the analysis of SSR markers in these species has helped identify potential molecular markers for species-level identification in genus *Fumaria*.

## 4.2. The complete chloroplast genomes of *Corydalis*

In this study, we report 36 new complete chloroplast genomes in Papaveraceae, including those of 25 species of *Corydalis*, which we sequenced, and 11 genomes that we assembled using sequences downloaded from the SRA database. Various structures have been reported to exist in the chloroplast genome of Papaveraceae [18, 23, 62], and our results confirmed the existence of more structures.

The average size of the chloroplast genome in subfamily Fumarioideae was 182,552 bp. The genomes contained an average of 75 CDSs, 30 tRNA genes, and 4 rRNA genes. In general, the *accD* gene was pseudogenized (*C. heracleifolia* and *C. semenowii*; Type 2) or lost in Papaveraceae. The *accD* gene encodes an acetyl-CoA carboxylase subunit. It regulates carbon flow into the fatty acid biosynthetic pathway [63] and is essential for leaf development in angiosperms [64]. This gene was known to be lost in the order Poales [65] and in the families Acoraceae [66], Rafflesiaceae [67], Geraniaceae [68], Fabaceae [69], Campanulaceae [63], and Oleaceae [28], and our results confirm that it has also been lost in family Papaveraceae.

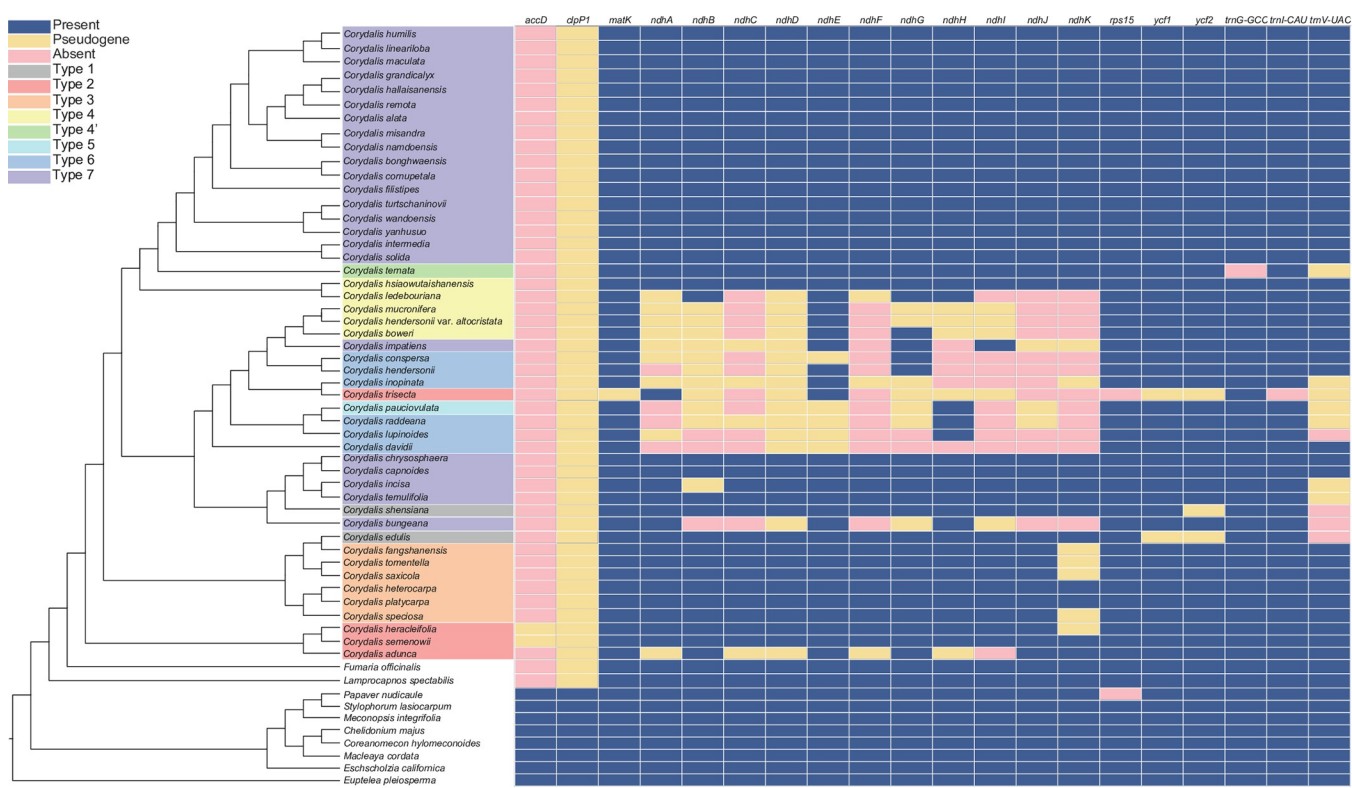

**Fig 9. Distribution patterns of gene loss in Papaveraceae.** The blue, red, and yellow blocks indicate the presence, pseudogenization, and absence of each gene, respectively. Genes are lost or pseudogenized near the inversion and rearrangement regions.

The NAD(P)H-dehydrogenase-like (NDH) complex is located on the thylakoid membrane of the chloroplast and plays an important role in mediating cyclic electron transport around photosystem I and promoting chloro-respiration [70]. The pseudogenization and/or loss of *ndhJ*, *ndhK*, and *ndhC* may have been associated with adjacent rearrangement events in the LSC region [23]. This pseudogenization was clearly confirmed by the results of our study. In particular, the pseudogenization of three genes were clearly detected in types 4, 5, and 6 of *Corydalis*, where the *rbcL–trnV-UAC* region of the LSC was rearranged. Moreover, pseudogenization was also confirmed in two species of Type 2 and two species of Type 7 *Corydalis*. In the *ndhK* gene of Type 3 *Corydalis* (three species), pseudogenization occurred due to the insertion of a 9 bp sequence (TCCTTTTTT). The *ndh* genes present in the IR and SSC regions were also lost with the pseudogene due to inversion and IR extension. In this study, we newly identified other pseudogenes of the *ndh* family. Part of the intron and exon 2 of the *ndhB* gene were lost due to the reversal event. This phenomenon was found in all species of types 5 and 6, and complete deletion was confirmed in some species (*C. lupinoides* and *C. davidii*, Type 6; and *C. bungeana*, Type 7). In *ndhA*, pseudogenization occurred in most species of Type 4, and complete deletion of this gene was confirmed in types 5 and 6 (accompanied by reversal events in the region from *ndhG* to *ycf1*). In addition, we found that some *ndh* genes (such as *ndhD* and *ndhH*) may have been lost randomly regardless of genome rearrangement (Fig 9). The loss and pseudogenization of *ndh* genes are known to occur frequently in heterotrophic plants, but have also been consistently identified in gymnosperms, such as conifers [10, 13] and Gnetales [71], and angiosperms such as Circaeasteraceae [72] and Orchidaceae [70]. Further studies are needed to assess whether the *ndh* genes in *Corydalis* spp. Were translocated to the nucleus or whether their loss represents a complete loss of the NDH complex.

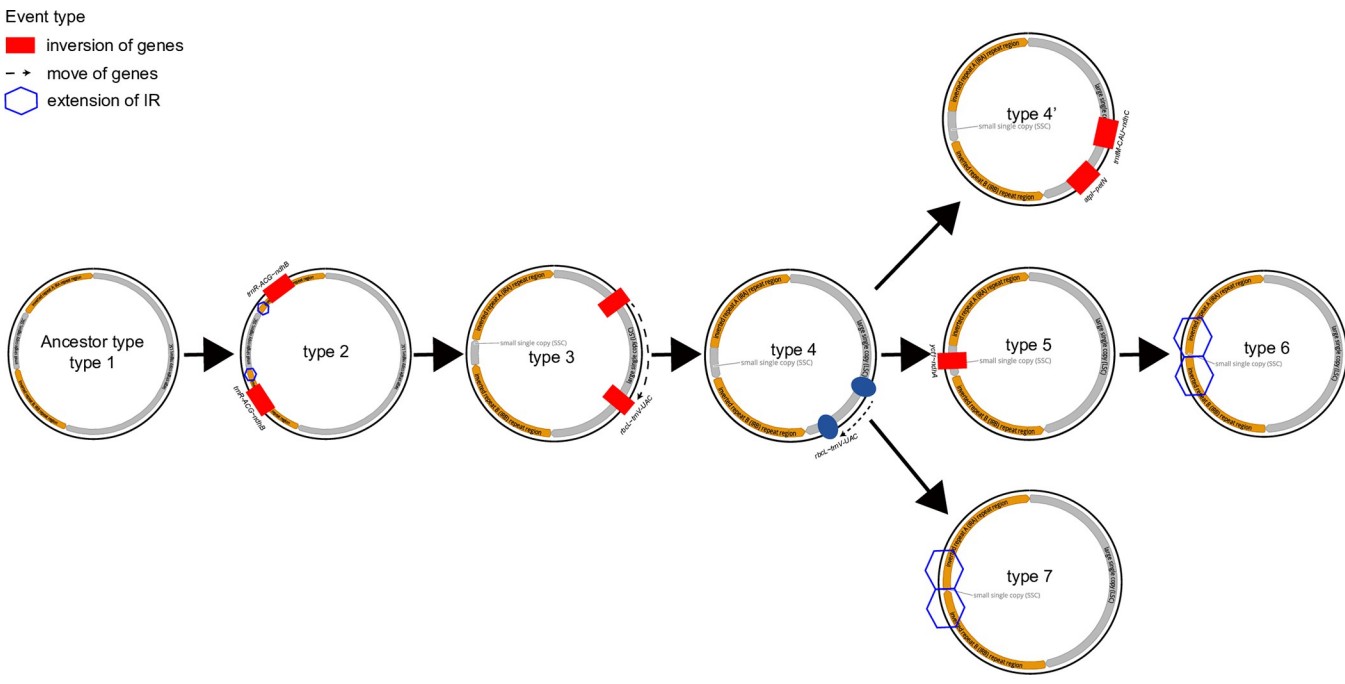

**Fig 10. Proposed model for the structural evolution of the chloroplast genome of *Corydalis*.**

Highly polymorphic and co-dominant loci such as SSRs have been used as molecular markers for population genetic studies and phylogenetic investigations [73]. In addition, repetitive sequences are also considered an important factor affecting gene replication, expansion, and genomic rearrangement [74]. A total of 34–94 SSRs and 49 long repetitive sequences were identified in this study. The identified SSRs had a high A/T content, and most (95.7%) mononucleotide repeats consisted of A/T bases. Thus, the SSRs reported here can be used as potential molecular markers for future studies on *Corydalis* species. In addition, large variations in long repeats in closely related species can reflect some degree of evolutionary flexibility [28, 75]. The long repeats identified in this study were longer than those reported in other angiosperms. Long repeats are typically < 100 bp in length; however, of the repeats found in *Corydalis* and *Fumaria*, 1,373 (57.2%) were 100–199 bp in length and 22 (0.9%) were ≥ 1,000 bp in length. Repeat-mediated genome instability often provides hotspots of genome rearrangement and evolutionary innovation [76, 77]. Recent studies have proposed hypotheses related to repeat-mediated structural variation in the chloroplast sequence of *Asarum* [24]. The various types of *Corydalis* identified in this study may also have emerged due to increased instability caused by repeat sequences.

Polymorphic regions can provide a wealth of valuable information for the development of DNA barcodes, and have been used as molecular markers in numerous phylogenetic studies. Here, we identified five regions with high variability in the chloroplast genome of *Corydalis*. The *clpP1* gene had the highest diversity ($Pi = 0.24837$), followed by *rps18* ($Pi = 0.10709$), *rps16* ($Pi = 0.09449$), *rpl22* ($Pi = 0.08193$), *ycf1* ($Pi = 0.07243$), and *rps11* ($Pi = 0.07184$). The diversity of the *clpP1* gene was 2.68 times higher than that of its intron ($Pi = 0.09274$); this may be because the exon region was lost and only the intron region was retained. Intron 2 was deleted in *C. trisecta*, and part of intron 1 was deleted in *C. edulis*. These results suggest that some taxa in subfamily Fumarioideae (along with *Lamprocapnos* and *Fumaria*) lost the exon of *clpP1*, similar to the loss of the *accD* gene. Further research is needed in this regard.

The consensus trees obtained using ML and BI methods were perfectly matched; however, the BI-based phylogeny had a higher resolution. *Corydalis* spp. Formed a clade with strong support based on the CDSs of the chloroplast genome (BS, PP = 100%, 1.00 each). A single lineage of *Corydalis* was similarly deduced from previous studies [23]. Within *Corydalis*, subgenus *Cremnocapnos* was the most divergent, and all taxa within this subgenus shared the structure of the Type 2 genome. Within *Cremnocapnos*, *C. abunca* diverged first, with *C. semenowii* and *C. heracleifloria* forming a sister group (BS, PP = 92%, 1.00 each). Subgenus *Sophorocapnos* diverged next (BS, PP = 100%, 1.00 each) and formed two major clades: *C. speciose* and the *C. platycarpa–C. heteocarpa* group (BS, PP = 100%, 1.00 each); and the *C. tomentella–C. speciose* and *C. fangshanensis–C. edulis* groups (BS, PP = 100%, 1.00 each). The clade containing *C. speciose* shared a common feature in that the position of the *rpl23* gene was shifted between *ndhF* and *rpl32*. This subgenus shared the structure of the Type 3 genome, with the exception of *C. edulis* (Type 1). Subgenus *Corydalis* was largely divided into two clades (BS, PP = 100%, 1.00 each), which can be morphologically divided into tubers (A) and roots (B). The B clade was further divided into two groups, including the clade with the most diverse genome structure. It was divided into one group with the Type 7 genome (except *C. shensiana*; Type 1) and one group comprising genome structure types 4, 5, and 6 (Type 2 for *C. trisecta* and Type 7 for *C. impatiens*). In the clade with tubers, *C. ledebouriana* and *C. hsiaowutaishanensis* (both Type 4) branched first, followed by *C. specios* (Type 4') and species with Type 7 genome structure. Most of the species endemic to Korea were located in this group.

Consistent with previous studies, our results indicated that the genus *Corydalis* is divided into three subgenera [23, 36]. However, because our sampling efforts were centered on the taxa distributed in South Korea, it was not possible to distinguish the sequences at the level of the tribe (a taxonomic level within the subgenus). A comprehensive study involving more taxa is needed for tribe-level phylogenetic analysis. Nevertheless, the high-resolution chloroplast genome sequences provided in our study may provide a promising resource for further elucidating the phylogeny and evolution of Papaveraceae (including genus *Corydalis*).

## 4.3. Evolution of the chloroplast structure of *Corydalis* through rearrangement of the chloroplast genome

The genome types reported were compared to identify rearrangement events in the chloroplast genome of *Corydalis* spp. Based on our findings, we propose a plastome rearrangement model for *Corydalis* that accounts for IR boundary shifts, inversions, and relocations (Fig 2). We also present a schematic diagram showing the events that likely occurred for each genome type (Fig 10). Type 1 was the ancestral *Corydalis* chloroplast genome and showed a chloroplast genome structure typical of angiosperms [61, 78]. In Type 2, an inversion event occurred in the region from *trnR-ACG* to *ndhB* in the IR region, accompanied by a slight expansion (S11 Fig). In Type 3, inversion and migration events occurred in the region from *rbcL* to *trnV-UAC* in the LSC region (S12 Fig). In Type 4, a movement event occurred once again in the LSC region from *rbcL* to *trnV-UAC*. Our findings suggest that the Type 4 structure diverged into three other types. In Type 4', two reversal events occurred in the LSC region (*atpI* to *petN* and *trnfM-CAU* to *ndhC*; S13 Fig); in Type 5, the region from *ycf1* to *ndhA* was reversed in the SSC region (S14 Fig); in Type 7, the IR region was extended to *ycf1*, and the SSC region became extremely short (S15 Fig). Type 6 was derived from Type 5 such that the IR region extended to *nahH*, resulting in an extremely short SSC region. To date, the reversal of the *ycf1–ndhA* region was thought to be caused by the sequential expansion of the Irb and Ira [23]. However, the Type 5 chloroplast genome of *Corydalis* identified in our study suggests that the reversal in the SSC region occurred first and was followed by the expansion of the Irb.

## 5. Conclusions

Elucidating and interpreting the evolutionary history of the chloroplast genome can help identify various structural variations in plant species such as *Corydalis* spp. The structural features previously identified in some chloroplast genomes were meticulously subdivided based on our findings. Although the structural features of chloroplasts are major phylogenetic features in some subgenera, those in subgenus *Corydalis* do not appear to follow a phylogenetic trend. However, further research on the chloroplast genomes of unexplored taxa can reveal new structural variations. Our analysis of the structural properties in the chloroplast genome was effective for inferring the phylogeny of genus *Corydalis*. However, further studies into the nuclear genome are still needed. We showed that IR extension and gene rearrangement play important roles in the evolution of the *Corydalis* chloroplast genome. We also found that subgenus *Corydalis*—which is endemic to the Korean Peninsula—was highly differentiated, but shared one type of chloroplast structure. This suggested that the various species may have differentiated very recently. More extensive sampling and studies focused on target species will provide important insights into the structural evolution of the chloroplast genome in Papaveraceae.

## Supporting information

**S1 Fig. Intron loss in the *rpl16* and *rps16* genes in *Corydalis ternate*.** Alignments are shown for the A. *rpl16* and B. *rps16* genes across various *Corydalis* spp.
(TIF)

**S2 Fig. Map of the chloroplast genome of *Fumaria officinalis*.** Genes shown outside the circle are transcribed in the counter counter-clockwise direction, and those inside the circle are transcribed in the clockwise direction. The colored bars indicate genes belonging to different functional groups. The inner circles denote the GC content (dark grey) and AT content (light grey) of the genome. The ψ signifies pseudogenes.
(TIF)

**S3 Fig. Type 1 of the chloroplast genome of *Corydalis* species.** Genes shown outside the circle are transcribed in the counter counter-clockwise direction, and those inside the circle are transcribed in the clockwise direction. The colored bars indicate genes belonging to different functional groups. The inner circles denote the GC content (dark grey) and AT content (light grey) of the genome. The ψ signifies pseudogenes.
(TIF)

**S4 Fig. Type 2 of the chloroplast genome of *Corydalis* species.** Genes shown outside the circle are transcribed in the counter counter-clockwise direction, and those inside the circle are transcribed in the clockwise direction. The colored bars indicate genes belonging to different functional groups. The inner circles denote the GC content (dark grey) and AT content (light grey) of the genome. The ψ signifies pseudogenes.
(TIF)

**S5 Fig. Type 3 of the chloroplast genome of *Corydalis* species.** Genes shown outside the circle are transcribed in the counter counter-clockwise direction, and those inside the circle are transcribed in the clockwise direction. The colored bars indicate genes belonging to different functional groups. The inner circles denote the GC content (dark grey) and AT content (light grey) of the genome. The ψ signifies pseudogenes.
(TIF)

**S6 Fig. Type 4 of the chloroplast genome of *Corydalis* species.** Genes shown outside the circle are transcribed in the counter counter-clockwise direction, and those inside the circle are transcribed in the clockwise direction. The colored bars indicate genes belonging to different functional groups. The inner circles denote the GC content (dark grey) and AT content (light grey) of the genome. The ψ signifies pseudogenes.
(TIF)

**S7 Fig. Type 4' of the chloroplast genome of *Corydalis* species.** Genes shown outside the circle are transcribed in the counter counter-clockwise direction, and those inside the circle are transcribed in the clockwise direction. The colored bars indicate genes belonging to different functional groups. The inner circles denote the GC content (dark grey) and AT content (light grey) of the genome. The ψ signifies pseudogenes.
(TIF)

**S8 Fig. Type 5 of the chloroplast genome of *Corydalis* species.** Genes shown outside the circle are transcribed in the counter counter-clockwise direction, and those inside the circle are transcribed in the clockwise direction. The colored bars indicate genes belonging to different functional groups. The inner circles denote the GC content (dark grey) and AT content (light grey) of the genome. The ψ signifies pseudogenes.
(TIF)

**S9 Fig. Type 6 of the chloroplast genome of *Corydalis* species.** Genes shown outside the circle are transcribed in the counter counter-clockwise direction, and those inside the circle are transcribed in the clockwise direction. The colored bars indicate genes belonging to different functional groups. The inner circles denote the GC content (dark grey) and AT content (light grey) of the genome. The ψ signifies pseudogenes.
(TIF)

**S10 Fig. Type 7 of the chloroplast genome of *Corydalis* species.** Genes shown outside the circle are transcribed in the counter counter-clockwise direction, and those inside the circle are transcribed in the clockwise direction. The colored bars indicate genes belonging to different functional groups. The inner circles denote the GC content (dark grey) and AT content (light grey) of the genome. The ψ signifies pseudogenes.
(TIF)

**S11 Fig. Sequence identity plots of the chloroplast genome sequences of *Corydalis* species (types 1 and 2), as aligned by the Shuffle-LAGAN algorithm using the *C. semenowii* sequence as a reference.** Grey arrows indicate the orientation of genes, red bars represent non-coding sequences, purple bars represent exons, and blue bars represent introns. The vertical scale indicates the percentage identity within 50–100%.
(TIF)

**S12 Fig. Sequence identity plots of the chloroplast genome sequences of *Corydalis* species (type 3), as aligned by the Shuffle-LAGAN algorithm using the *C. speciose* sequence as a reference.** Grey arrows indicate the orientation of genes, red bars represent non-coding sequences, purple bars represent exons, and blue bars represent introns. The vertical scale indicates the percentage identity within 50–100%.
(TIF)

**S13 Fig. Sequence identity plots of the chloroplast genome sequences of *Corydalis* species (types 4 and 4'), as aligned by the Shuffle-LAGAN algorithm using the *C. boweri* sequence as a reference.** Grey arrows indicate the orientation of genes, red bars represent non-coding

sequences, purple bars represent exons, and blue bars represent introns. The vertical scale indicates the percentage identity within 50–100%.
(TIF)

**S14 Fig. Sequence identity plots of the chloroplast genome sequences of *Corydalis* species (types 5 and 6), as aligned by the Shuffle-LAGAN algorithm using the *C. raddeana* sequence as a reference.** Grey arrows indicate the orientation of genes, red bars represent non-coding sequences, purple bars represent exons, and blue bars represent introns. The vertical scale indicates the percentage identity within 50–100%.
(TIF)

**S15 Fig. Sequence identity plots of the chloroplast genome sequences of *Corydalis* species (type 7), as aligned by the Shuffle-LAGAN algorithm using the *C. alata* sequence as a reference.** Grey arrows indicate the orientation of genes, red bars represent non-coding sequences, purple bars represent exons, and blue bars represent introns. The vertical scale indicates the percentage identity within 50–100%.
(TIF)

**S1 Table. Information on the collected 26 species.**
(XLSX)

**S2 Table. Information on 11 species of *Corydalis* registered with NCBI SRA.**
(XLSX)

**S3 Table. Information on 22 species down-loaded and used from NCBI nucleotide.**
(XLSX)

**S4 Table. Information from RSCU on 48 species of *Corydalis*.**
(XLSX)

**S5 Table. Information on the type and number of simple sequence repeats (SSRs) for 48 species of *Corydalis* and 1 species of *Fumaria*.**
(XLSX)

**S6 Table. Information from SSR on 48 species of *Corydalis* and 1 species of *Fumaria*.**
(XLSX)

**S7 Table. Types and numbers of long repeats in the chloroplast genomes of 48 species of *Corydalis* and 1 species of *Fumaria* by using REPuter.**
(XLSX)

## Acknowledgments

We thank Sa-Bum Jang, Kang Hyup Lee, Jung Sim Lee, Myoung Ja Nam, and Eun-Ho Lee for tissue sampling and laboratory assistance throughout the project.

## Author Contributions

**Data curation:** Sang-Chul Kim, Young-Ho Ha.

**Formal analysis:** Sang-Chul Kim.

**Funding acquisition:** Hyuk-Jin Kim.

**Investigation:** Sang-Chul Kim, Young-Ho Ha, Beom Kyun Park, Ju Eun Jang, Eun Su Kang, Young-Soo Kim, Tae-Hee Kimspe, Hyuk-Jin Kim.

**Methodology:** Sang-Chul Kim.

**Project administration:** Sang-Chul Kim, Hyuk-Jin Kim.

**Validation:** Young-Soo Kim, Tae-Hee Kimspe.

**Visualization:** Sang-Chul Kim, Beom Kyun Park, Ju Eun Jang, Eun Su Kang.

**Writing – original draft:** Sang-Chul Kim, Young-Ho Ha, Beom Kyun Park, Ju Eun Jang, Eun Su Kang, Young-Soo Kim, Tae-Hee Kimspe.

**Writing – review & editing:** Sang-Chul Kim, Young-Ho Ha, Hyuk-Jin Kim.

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
