## [Decision Letter · Decision Letter 0]

13 Jun 2023

PONE-D-23-03735Comparative analysis of the complete chloroplast genome of Papaveraceae to identify rearrangements within the Corydalis chloroplast genomePLOS ONE

Dear Dr. Kim,

Thank you for submitting your manuscript to PLOS ONE. After careful consideration, we feel that it has merit but does not fully meet PLOS ONE’s publication criteria as it currently stands. Therefore, we invite you to submit a revised version of the manuscript that addresses the points raised during the review process.

We look forward to receiving your revised manuscript.

Kind regards,

Tzen-Yuh Chiang

Academic Editor

PLOS ONE

6. We note that Figure 8 in your submission contain copyrighted images. All PLOS content is published under the Creative Commons Attribution License (CC BY 4.0), which means that the manuscript, images, and Supporting Information files will be freely available online, and any third party is permitted to access, download, copy, distribute, and use these materials in any way, even commercially, with proper attribution. For more information, see our copyright guidelines: http://journals.plos.org/plosone/s/licenses-and-copyright.

a. You may seek permission from the original copyright holder of Figure 8 to publish the content specifically under the CC BY 4.0 license.

b.If you are unable to obtain permission from the original copyright holder to publish these figures under the CC BY 4.0 license or if the copyright holder’s requirements are incompatible with the CC BY 4.0 license, please either i) remove the figure or ii) supply a replacement figure that complies with the CC BY 4.0 license. Please check copyright information on all replacement figures and update the figure caption with source information. If applicable, please specify in the figure caption text when a figure is similar but not identical to the original image and is therefore for illustrative purposes only.

Reviewers' comments:

Reviewer's Responses to Questions

**Comments to the Author**

1. Is the manuscript technically sound, and do the data support the conclusions?

Reviewer #1: Partly

2. Has the statistical analysis been performed appropriately and rigorously? 

Reviewer #1: Yes

3. Have the authors made all data underlying the findings in their manuscript fully available?

Reviewer #1: Yes

4. Is the manuscript presented in an intelligible fashion and written in standard English?

Reviewer #1: Yes

5. Review Comments to the Author

Reviewer #1: PONE-D-23-03735

In this manuscript (PONE-D-23-03735), authors reported complete chloroplast genomes of 36 Corydalis spp. and one Fumaria species, and compared these genomes with 22 other taxa’s genomes. The study could be interesting for publication in PLOS ONE. However, the manuscript has some questions which I think need to be addressed.

Major：

1. From Figure 6, we knew some chloroplast genomes were incomplete. The authors should firstly confirm the analyzed 58 chloroplast genomes complete or incomplete. Then, compare the complete chloroplast genomes structure, gene contents, molecular evolution, and so on.

2.S4 Table and S5 Table should be in the manuscript, not in the supplementary section.

3.There were so many chloroplast genomes of Papaveraceae and Corydalis, divergence time estimation may be analyzed and discussed in details.

Minor:

1.There are some repeats of subheadings.

6. PLOS authors have the option to publish the peer review history of their article (what does this mean?). If published, this will include your full peer review and any attached files.

Reviewer #1: No

---

## [Author Response · Author response to Decision Letter 0]

22 Jun 2023

1. From Figure 6, we knew some chloroplast genomes were incomplete. The authors should firstly confirm the analyzed 58 chloroplast genomes complete or incomplete. Then, compare the complete chloroplast genomes structure, gene contents, molecular evolution, and so on. 

: All 58 chloroplast genomes we analyzed has complete structures. Figure 6 shows the rearrangement of the genome.

2. S4 Table and S5 Table should be in the manuscript, not in the supplementary section.

: We have included two tables in the manuscript based on the reviewer's comments.

3. There were so many chloroplast genomes of Papaveraceae and Corydalis, divergence time estimation may be analyzed and discussed in details.

: Our thesis was written based on the comparison of the chloroplast genome structure of corydalis, and the amount of papers increased when the branch estimation was analyzed, so the amount of papers was divided for future analysis.

Minor:

1. There are some repeats of subheadings. 

: Repetition of subheadings is acknowledged. However, a total of nine chloroplast structures were identified, and we tried to explain each one.

---

## [Decision Letter · Decision Letter 1]

24 Jul 2023

Comparative analysis of the complete chloroplast genome of Papaveraceae to identify rearrangements within the Corydalis chloroplast genome

PONE-D-23-03735R1

Dear Dr. Kim,

We’re pleased to inform you that your manuscript has been judged scientifically suitable for publication and will be formally accepted for publication once it meets all outstanding technical requirements.

Kind regards,

Tzen-Yuh Chiang

Academic Editor

PLOS ONE

Additional Editor Comments (optional):

Reviewers' comments:

Reviewer's Responses to Questions

**Comments to the Author**

1. If the authors have adequately addressed your comments raised in a previous round of review and you feel that this manuscript is now acceptable for publication, you may indicate that here to bypass the “Comments to the Author” section, enter your conflict of interest statement in the “Confidential to Editor” section, and submit your "Accept" recommendation.

Reviewer #1: All comments have been addressed

2. Is the manuscript technically sound, and do the data support the conclusions?

Reviewer #1: Yes

3. Has the statistical analysis been performed appropriately and rigorously? 

Reviewer #1: Yes

4. Have the authors made all data underlying the findings in their manuscript fully available?

Reviewer #1: Yes

5. Is the manuscript presented in an intelligible fashion and written in standard English?

Reviewer #1: Yes

6. Review Comments to the Author

Reviewer #1: This MS version has greatly improved;however, there are some places also needing revision, such as writing normal or italics styles, IRa, IRb, and so on. Please see details in the revised version of R1.

7. PLOS authors have the option to publish the peer review history of their article (what does this mean?). If published, this will include your full peer review and any attached files.

Reviewer #1: No

---

## [Editor Report · Acceptance letter]

15 Sep 2023

PONE-D-23-03735R1 

Comparative analysis of the complete chloroplast genome of Papaveraceae to identify rearrangements within the *Corydalis* chloroplast genome 

Dear Dr. Kim:

I'm pleased to inform you that your manuscript has been deemed suitable for publication in PLOS ONE. Congratulations! Your manuscript is now with our production department. 

Kind regards, 

on behalf of

Dr. Tzen-Yuh Chiang 

Academic Editor

PLOS ONE